# Structure analyses reveal a regulated oligomerization mechanism of the PlexinD1/GIPC/myosin VI complex

**Guijun Shang[1], Chad A Brautigam[2,3], Rui Chen[1], Defen Lu[4], Jesús Torres-Vázquez[5], Xuewu Zhang[1,2]\***

[1]Department of Pharmacology, University of Texas Southwestern Medical Center, Dallas, United States; [2]Department of Biophysics, University of Texas Southwestern Medical Center, Dallas, United States; [3]Department of Microbiology, University of Texas Southwestern Medical Center, Dallas, United States; [4]Department of Molecular Biology, University of Texas Southwestern Medical Center, Dallas, United States; [5]Department of Cell Biology, Skirball Institute of Biomolecular Medicine, New York, United States

**Abstract** The GIPC family adaptor proteins mediate endocytosis by tethering cargo proteins to the myosin VI motor. The structural mechanisms for the GIPC/cargo and GIPC/myosin VI interactions remained unclear. PlexinD1, a transmembrane receptor that regulates neuronal and cardiovascular development, is a cargo of GIPCs. GIPC-mediated endocytic trafficking regulates PlexinD1 signaling. Here, we unravel the mechanisms of the interactions among PlexinD1, GIPCs and myosin VI by a series of crystal structures of these proteins in apo or bound states. GIPC1 forms a domain-swapped dimer in an autoinhibited conformation that hinders binding of both PlexinD1 and myosin VI. PlexinD1 binding to GIPC1 releases the autoinhibition, promoting its interaction with myosin VI. GIPCs and myosin VI interact through two distinct interfaces and form an open-ended alternating array. Our data support that this alternating array underlies the oligomerization of the GIPC/Myosin VI complexes in solution and cells.

**\*For correspondence:** xuewu. zhang@utsouthwestern.edu

**Competing interests:** The authors declare that no competing interests exist.

## Introduction

GIPC1 (GAIP interacting protein, C-terminus 1) and its two paralogs GIPC2 and GIPC3 are universal adaptor proteins that bind and regulate vesicular trafficking of many transmembrane proteins (*Katoh, 2013*). Based on the cargo proteins it interacts with, GIPC1 is also named syndecan-4 binding protein (Synectin), Neuropilin-1-interacting protein (NIP), insulin-like growth factor-1 receptor interacting protein (IIP1), GLUT1 C-terminal binding protein (GLUT1CBP), M-SemF cytoplasmic domain-associated protein (SEMCAP-1) and Tax-interacting protein 2 (TIP2) (*Bunn et al., 1999*; *Cai and Reed, 1999*; *Gao et al., 2000*; *Ligensa et al., 2001*; *Rousset et al., 1998*; *Wang et al., 1999*). GIPCs contribute to the endocytosis process by tethering the cargo proteins on endocytic vesicles to the motor protein myosin VI (*Bunn et al., 1999*; *Naccache et al., 2006*; *Reed et al., 2005*). Myosin VI is the only known actin-based motor that walks toward the minus/pointed end of actin filaments and therefore is specialized at transporting endocytic vesicles inward from the cell cortex (*Spudich and Sivaramakrishnan, 2010*; *Sweeney and Houdusse, 2007*). Myosin VI and GIPCs also contribute to earlier steps of endocytosis, including facilitating the formation of clathrin-coated pits and clustering ligand-bound cell surface receptors into the pits (*Hasson, 2003*). Another function of the GIPC/myosin VI complex is to act as an anchor on the cell surface. This function is

critical for maintaining the structural integrity of stereocilia in the inner ear, and many mutations of GIPC3 or myosin VI have been found to cause familial hearing loss (*Katoh, 2013*).

GIPCs contain a central PDZ (PSD-95, Dlg and ZO-1) domain that mediates the interaction with the C-terminal PDZ-binding motif (PBM) in cargo proteins. While the general rules of specificity in PDZ/PBM interactions are well understood (*Ye and Zhang, 2013*), the mechanisms by which GIPCs distinguish their cargo proteins from many proteins containing similar PBMs were unclear. The PDZ domain in GIPCs is flanked by a N-terminal GIPC-homology 1 (GH1) domain and a C-terminal GH2 domain (*Katoh, 2013*). The GH1 and GH2 domains are unique to the GIPC family and lack obvious sequence similarity to other proteins. The GH1 and PDZ domains together mediate oligomerization of GIPCs (*Bunn et al., 1999*; *Gao et al., 2000*; *Jeanneteau et al., 2004*; *Kedlaya et al., 2006*; *Reed et al., 2005*). The mechanisms of the oligomerization and its role in the regulation of GIPCs remained unclear. The GH2 domain mediates the interaction with myosin VI (*Naccache et al., 2006*; *Reed et al., 2005*). Cargo binding to the PDZ domain in GIPC1 promotes the interaction with myosin VI, suggesting that the two binding events in GIPCs are regulated and communicate with each other through a steric or allosteric mechanism (*Naccache et al., 2006*).

Myosin VI contains a motor domain, an IQ motif domain, a coiled-coil region, and a C-terminal cargo-binding (CBD) domain. The motor activity of myosin VI is normally suppressed by autoinhibition, which can be released by binding of adaptor proteins through a mechanism that is poorly understood (*French et al., 2017*). An 'RRL' (single-letter amino acid code) motif between the coiled-coil and CBD in myosin VI is critical for binding to GIPCs (*He et al., 2016*; *Spudich et al., 2007*; *Wollscheid et al., 2016*), but the mode of the interaction was unknown. Processive walking of myosin VI along actin filaments requires its dimerization, mediated by both the coiled-coil and binding of adaptor proteins (*Masters and Buss, 2017*; *Park et al., 2006*; *Phichith et al., 2009*; *Spudich et al., 2007*; *Yu et al., 2009*). Larger oligomers of myosin VI have been implicated in coordinated long-distance runs over dense and interlaced actin networks (*Sivaramakrishnan and Spudich, 2009*). Hence, oligomerized GIPCs may play a role in both releasing the autoinhibition and promoting the oligomerization of myosin VI, but the underlying mechanisms were unclear.

The transmembrane receptor PlexinD1 contains a C-terminal 'SEA' motif that constitutes a PBM and interacts with GIPC1 (*Gay et al., 2011*). The plexin family proteins are the major receptors for the guidance molecules semaphorins. Semaphorin-activated plexin primarily transduces repulsive guidance signal that is critical for the development of the nervous and cardiovascular systems (*Tran et al., 2007*). Semaphorin binding induces the formation of the active dimer of plexin (*Janssen et al., 2010*; *Liu et al., 2010*; *Nogi et al., 2010*). The dimerization activates the GTPase-activating protein (GAP) domain in the plexin cytoplasmic region (referred to as plexin$_{cyto}$ thereafter), which transduces the signal by switching off the small GTPase Rap (*He et al., 2009*; *Wang et al., 2013*; *Wang et al., 2012*). Plexin$_{cyto}$ also contains a Juxtamembrane segment (JM-segment) and a Rho GTPase Binding Domain (RBD) that both play regulatory roles in signaling (*Bell et al., 2011*; *He et al., 2009*; *Tong et al., 2007*, *2009*). Semaphorin-stimulated endocytosis of plexin contributes to the repulsive guidance function (*Fournier et al., 2000*; *Tojima et al., 2010*). A recent study has shown that GIPC1-mediated endocytic trafficking and sorting of Sema3E-activated PlexinD1 play important regulatory roles in PlexinD1 signaling (*Burk et al., 2017*).

Structures of plexins, the PDZ domain of GIPC2 (PDB ID: 3GGE) and the GIPC-binding region in myosin VI have been reported previously (*Bell et al., 2011*; *He et al., 2016*, *2009*; *Tong et al., 2009*; *Wang et al., 2013*; *Wang et al., 2012*; *Wollscheid et al., 2016*). However, structures of full-length GIPCs and their complexes with either cargos or myosin VI are not available, hindering our understanding of the mechanisms underlying GIPC functions. In this study, we present comprehensive structural analyses of PlexinD1$_{cyto}$, GIPCs and myosin VI in the apo and bound states. The structures and associated mutational analyses elucidate the structural basis and regulation mechanisms for the interaction between PlexinD1 and GIPC1. In addition, the structures reveal an unanticipated mechanism for high-order oligomerization of GIPCs and myosin VI, which may underlie the clustering on the plasma membrane and efficient endocytic transport of PlexinD1 and other cargo proteins by GIPCs and myosin VI.

# Results

## Structure of GIPC1 in the dimeric and autoinhibited state

We crystallized the GH1-PDZ-GH2 domains of mouse GIPC1 and determined its X-ray crystal structure to 1.9 Å resolution (*Figure 1*, *Figure 1—figure supplement 1* and *Table 1*). The structure of the PDZ domain of GIPC1 is similar to that of the isolated PDZ domain of GIPC2 (PDB ID: 3GGE), featuring a typical PDZ fold with five β-strands and two α-helices (*Figure 1C*). The GH1 domain adopts an ubiquitin-like fold composed of four β-strands and one α-helix, although it lacks obvious sequence similarity to ubiquitin (*Figure 1C* and *Figure 1—figure supplement 2*). The GH2 domain forms a four-helix globular fold (*Figure 1C*).

A prominent feature of the GIPC1 structure is that two molecules in the asymmetric unit form a domain-swapped dimer (*Figure 1*). The domain swap is the result of an exchange of the last β-strand between the two GH1 domains (*Figure 1C*). The segment N-terminal to the last β-strand of one GH1 domain meets its counterpart of the other GH1 domain, making extensive intermolecular interactions at the dimer interface. The extension N-terminal to the GH1 domain also contributes to the

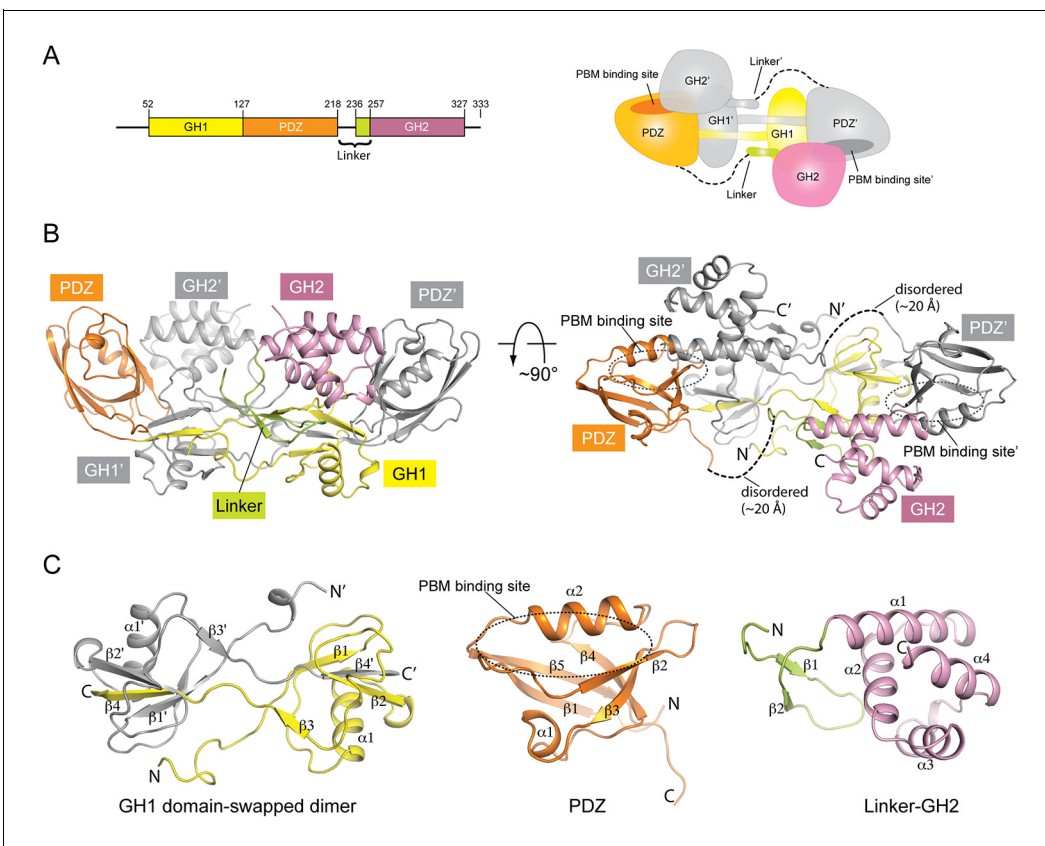

**Figure 1.** Crystal structure of mouse GIPC1 in the autoinhibited conformation. (**A**) Domain structure of mouse GIPC1 and cartoon representation of the overall architecture of the domain-swapped dimer. The color scheme for one subunit in the dimer of the cartoon representation is the same as in the domain structure. The other subunit is colored gray. (**B**) Two orthogonal views of the GIPC1 structure. The dotted lines indicate the disordered portion of the linker between the PDZ and GH2 domains. The color scheme is the same as in (**A**). (**C**) Expanded views of the individual domains.

The following figure supplements are available for figure 1:

**Figure supplement 1.** Representative electron density of the apo-GIPC1 structure.

**Figure supplement 2.** Sequence alignment of GIPC1, 2 and 3.

**Table 1.** Data collection and refinement statistics.

| | GIPC1 | PlexinD1 | PlexinD1/ GIPC1 complex | GIPC1-GH2/ myosin VI-HCBD complex | GIPC2-GH2/ myosin VI-HCBD complex |
|---|---|---|---|---|---|
| **Data collection** | | | | | |
| Space group | $P2_1$ | $P2_1$ | $P6_122$ | I222 | C2 |
| **Cell dimensions** | | | | | |
| a, b, c (Å) | 45.4, 77.6, 80.3 | 69.9, 164.5, 84.3 | 99.8, 99.8, 531.9 | 115.0, 164.1, 171.8 | 171.8, 53.2, 122.8 |
| $\alpha$, $\beta$, $\gamma$ (°) | 90.00, 89.95, 90.00 | 90.00, 99.47, 90.00 | 90.00, 90.00, 120.00 | 90.00, 90.00, 90.00 | 90.00 108.23 90.00 |
| Content of the asymmetric unit | GIPC1 dimer, 1 | PlexinD1, 2 | PlexinD1, 1; GIPC1, 1 | GH2, 5; HCBD, 5 | GH2, 5; HCBD, 5 |
| Resolution (Å) | 50.0–1.90 (1.93–1.90)* | 50.0–2.70 (2.75–2.70) | 50.0–3.20 (3.26–3.20) | 50.0–3.5 (3.56–3.50) | 50–3.6 (3.66–3.60) |
| $R_{sym}$ (%) | 6.2 (56.1) | 7.2 (38.8) | 14.1 (81.9) | 15.7 (>100) | 15.9 (47.2) |
| $R_{pim}$ (%) | 3.4 (31.9) | 4.2 (23.1) | 4.1 (36.8) | 7.4 (49.3) | 7.9 (27.2) |
| $I/\sigma$ | 21.3 (2.1) | 29.9 (3.4) | 17.5 (2.0) | 10.3 (1.6) | 11.0 (2.7) |
| $CC_{1/2}$# | 0.782 | 0.922 | 0.679 | 0.653 | 0.859 |
| Completeness (%) | 100.0 (100.0) | 99.4 (96.8) | 100.0 (100.0) | 98.1 (97.0) | 98.7 (95.4) |
| Redundancy | 4.1 (4.0) | 3.9 (3.6) | 12.0 (5.6) | 6.1 (6.0) | 4.7 (3.7) |
| **Refinement** | | | | | |
| Resolution (Å) | 40.1–1.9 (1.95–1.90) | 37.1–2.7 (2.77–2.70) | 48.0–3.2 (3.28–3.20) | 47.1–3.5 (3.60–3.50) | 42.9–3.6 (3.79–3.60) |
| No. reflections | 40623 | 47171 | 26781 | 18708 | 9763 |
| $R_{work}/R_{free}$ (%) | 13.6 (19.4)/17.5 (24.8) | 21.5 (31.8)/25.6 (36.9) | 17.9 (29.3)/21.6 (32.4) | 18.7 (26.8)/24.0 (32.3) | 21.3 (26.6)/27.8 (37.6) |
| No. atoms | | | | | |
| Protein | 4120 | 8092 | 5416 | 4858 | 4590 |
| Ligand/ion | 0 | 0 | 35 | 0 | 0 |
| Water | 645 | 98 | 19 | 0 | 0 |
| B-factors | | | | | |
| Protein | 21.9 | 80.0 | 57.8 | 70.4 | 57.8 |
| Ligand/ion | | | 115.2 | | |
| Water | 33.5 | 61.0 | 25.7 | | |
| R.m.s deviations | | | | | |
| Bond lengths (Å) | 0.006 | 0.003 | 0.005 | 0.007 | 0.003 |
| Bond angles (°) | 0.9276 | 0.617 | 0.834 | 0.823 | 0.603 |
| Ramanchandran plot | | | | | |
| Favored (%) | 96.5 | 96.2 | 96.7 | 94.1 | 96.2 |
| Allowed (%) | 3.5 | 3.6 | 3.2 | 5.9 | 3.7 |
| Disallowed (%) | 0 | 0.2 | 0.1 | 0 | 0.2 |

*Numbers in parenthesis are for the highest resolution shell.
# $CC_{1/2}$ values shown are for the highest resolution shell.

dimer interface. Due to the domain swap, the PDZ domain from one molecule is located in close proximity to the GH1 domain of the dimer partner. The two domains form an integrated structural module through extensive interactions, further stabilizing the architecture of the GIPC1 dimer. The overall shape of the dimeric GH1-PDZ module resembles a canoe with the two GH1 domains at the middle and the two PDZ domains curving upwards at each end. The PBM-binding groove between helix $\alpha2$ and strand $\beta2$ in the PDZ domain is placed at the concave upper face of the canoe-like dimer (*Figure 1*). This dimeric configuration provides a structural basis for the previously reported self-association of GIPC1 (*Bunn et al., 1999*; *Gao et al., 2000*; *Jeanneteau et al., 2004*; *Kedlaya et al., 2006*; *Reed et al., 2005*). The high degree of sequence similarity in the GIPC family

suggests that GIPC2 and GIPC3 also form similar domain-swapped dimers (*Figure 1—figure supplement 2*).

The C-terminal portion (residues 236–257) of the PDZ-GH2 linker is well structured and contains two short β-strands, which together with a small segment in GH1 forms a small three-stranded β-sheet (*Figures 1* and *2A*). However, the N-terminal portion (residues 219–235) of the linker is disordered, making the connection between the PDZ and linker of the two subunits in the dimer ambiguous. We tentatively assigned the connection as shown in *Figure 1A and B*, in which the span of the disordered region in the linker is ~20 Å. Based on this assignment, one linker-GH2 packs against the GH1 domain from the same subunit as well as the PDZ domain from the other subunit. The alternative connection is less likely but possible, as the gap of ~40 Å in this configuration can be spanned by the 17-residue disordered linker region. The GH2 domain wedges itself between the PDZ-GH2 linker and the PDZ domain, reminiscent of a passenger sitting in the canoe formed by the dimeric GH1-PDZ modules. Notably, the loop between helices α1 and α2 in the GH2 domain partially occupies the PBM-binding groove in the PDZ domain (*Figure 2B*). Conversely, the PDZ domain blocks one face on the GH2 domain that binds myosin VI (as defined by the structures of the GH2/myosin

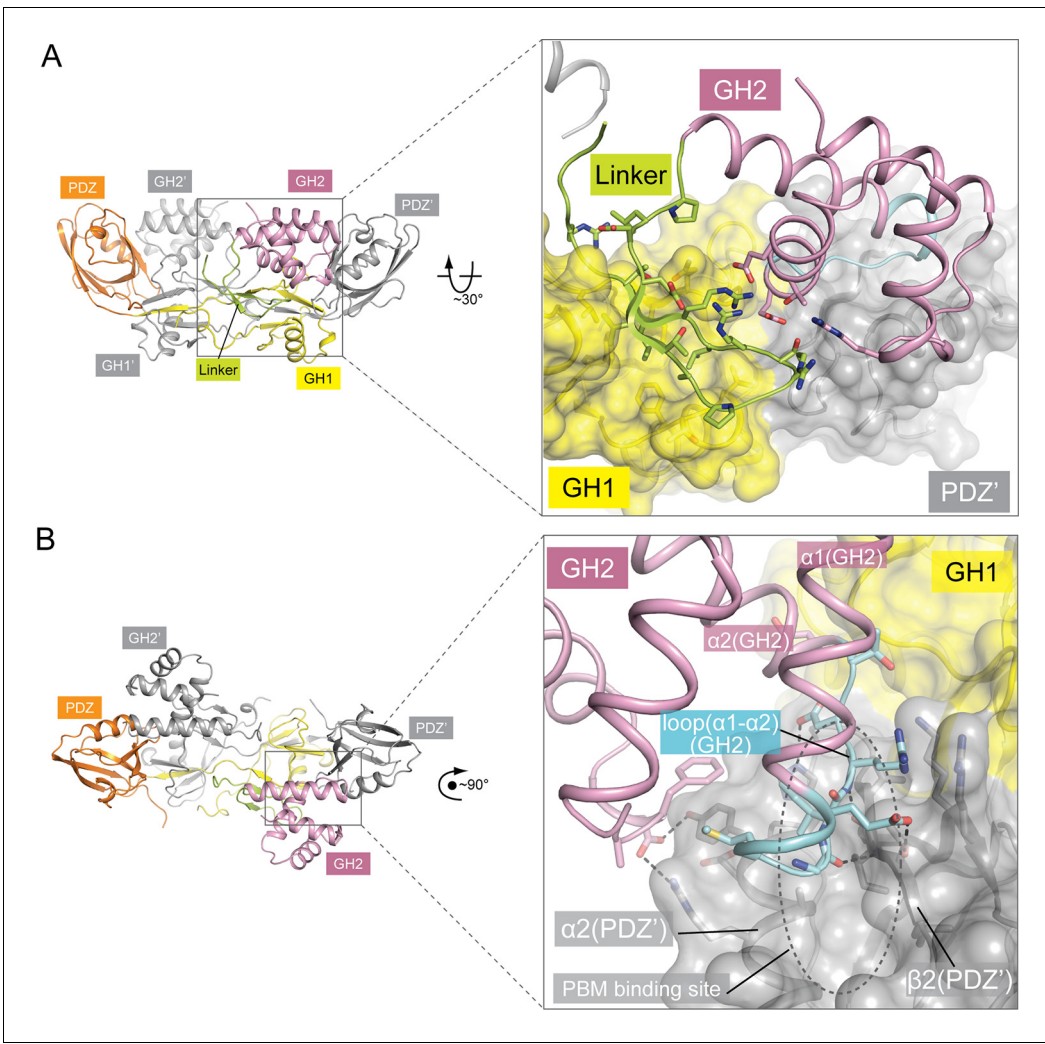

**Figure 2.** Autoinhibitory interactions mediated by the linker-GH2 domains in the apo-GIPC1 structure. (**A**) Interactions of the linker with the GH1 and GH2 domains. (**B**) Interaction between the GH2 and PDZ domains. The loop between helices α1 and α2 in the GH2 domain that partially blocks the PBM binding site is labeled 'loop(α1-α2)' and highlighted in cyan.

VI complexes below). The apo-GIPC1 structure therefore represents an autoinhibited conformation that hinders binding to both PBM-containing cargo proteins and myosin VI.

## Structure of the PlexinD1/GIPC1 complex

The JM segment in plexin$_{cyto}$ is relatively flexible and can adopt multiple conformations for regulating plexin activity (**Bell et al., 2011**; **He et al., 2009**; **Pascoe et al., 2015**; **Tong et al., 2009**; **Wang et al., 2013**). While full-length PlexinD1$_{cyto}$ failed to crystallize, we crystallized and determined the structures of a construct with the JM segment truncated (still referred to as PlexinD1$_{cyto}$ for simplicity), both in the apo-form (to 2.7 Å resolution) and in complex with GIPC1 (to 3.2 Å resolution) (**Figure 3**, **Figure 3—figure supplement 1** and **Table 1**). The overall structure of apo-PlexinD1$_{cyto}$ are similar to other plexin$_{cyto}$ structures reported previously (**Figure 3B**) (**Bell et al., 2011**; **He et al., 2009**; **Pascoe et al., 2015**; **Tong et al., 2009**; **Wang et al., 2013**). The RBD forms an independent ubiquitin-like domain that packs against one side of the GAP domain. The two GAP-homology regions (C1 and C2) fold together to form the GAP domain (**Figure 3A and B**). The GAP domain of plexins contains a regulatory element referred to as the activation segment, which toggles between the closed inactive and open active conformations under the control of plexin dimerization (**Wang et al., 2013**). The activation segment in the PlexinD1$_{cyto}$ structure appears partially open (**Figure 3B**), although it was crystallized in the monomeric state. The relatively high basal GAP activity of PlexinD1 may be due to this propensity of the activation segment to adopt the open conformation (**Wang et al., 2012**).

The structure of PlexinD1$_{cyto}$ in complex with GIPC1 is similar to apo-PlexinD1$_{cyto}$, except in the C-terminal tail that binds GIPC1 (**Figure 3B** and **Figure 3—figure supplement 1**; see details below). The entire PDZ-GH2 linker and GH2 domain of GIPC1 in the PlexinD1$_{cyto}$/GIPC1 complex are invisible in the electron-density map. A gel analysis shows that the GIPC1 protein is intact in crystals (**Figure 3—figure supplement 2**), suggesting that the linker-GH2 domains are present but completely disordered in the PlexinD1/GIPC1 complex structure. As mentioned above, the PDZ and GH2 domains mutually inhibit each other in the autoinhibited apo-structure of GIPC1. Binding of PlexinD1 to GIPC1 is expected to dislodge the GH2 domain from its autoinhibitory interaction with the PDZ domain, rendering the GH2 domain and the PDZ-GH2 linker conformationally flexible in relation to the GH1-PDZ module (**Figure 3C**). As a result, the GH2 domain can more readily interact with myosin VI, which explains the enhanced recruitment of myosin VI by GIPC1 when GIPC1 is bound to cargo proteins as reported previously (**Naccache et al., 2006**).

The GH1 and PDZ domains of GIPC1 in the PlexinD1/GIPC1 complex are well structured and form the same domain-swapped dimer as apo-GIPC1 (**Figure 3C**). The curvature of the canoe-shaped dimer is, however, different in the two structures, suggesting that the dimer interface between the two GH1 domains has a certain degree of plasticity (**Figure 3—figure supplement 3**). The variation of the dimer interface in the PlexinD1$_{cyto}$/GIPC1 complex structure could result from the loss of the interfacial interactions made by the PDZ-GH2 linker present in the apo-GIPC1 structure. The two PlexinD1$_{cyto}$ molecules are placed at the opposite sides of the GIPC1 dimer and do not contact each other (**Figure 3C**). Simple modeling shows that the 2:2 PlexinD1/GIPC1 complex is compatible with full-length PlexinD1 in lipid membrane interacting with GIPC1 in the cytosol, which can use its flexibly attached GH2 domain to recruit myosin VI (**Figure 3C**).

In the apo-PlexinD1$_{cyto}$ structure, the C-terminal tail region (residues 1916–1925) constitutes the C-terminal part of the last helix in the GAP domain, except for the last two residues (Glu1924 and Ala1925) which are disordered (**Figure 3B** and **Figure 3—figure supplement 1**). In contrast, the tail segment in the PlexinD1$_{cyto}$/GIPC1 complex adopts an ordered, extended conformation (**Figure 3B** and **Figure 3—figure supplement 1**). As expected, the PBM interacts with the PDZ domain in GIPC1 in the typical PBM/PDZ-binding mode, where the PBM forms a short $\beta$-strand to pack in an anti-parallel fashion with strand $\beta2$ in the PDZ domain (**Figure 4A**) (**Ye and Zhang, 2013**). The side-chain of the very C-terminal residue (Ala1925) in the PBM is buried in the hydrophobic pocket between helix $\alpha2$ and strand $\beta2$ of the PDZ domain, while its carbonyl group interacts with the backbone atoms of Leu143 and Gly144 in the PDZ domain. Ser1923 and Glu1924 in the PBM contribute to the specific binding by interacting with His191 and Thr146/Arg159 respectively in the PDZ domain. In addition to these canonical PBM/PDZ interactions, GIPC1 makes extensive contacts with other residues in the C-terminal tail as well as the two preceding helices of PlexinD1, resulting in a large interface with the total buried surface area of 2072 Å$^2$ (**Figure 4A**). Notably, Ile1918 and

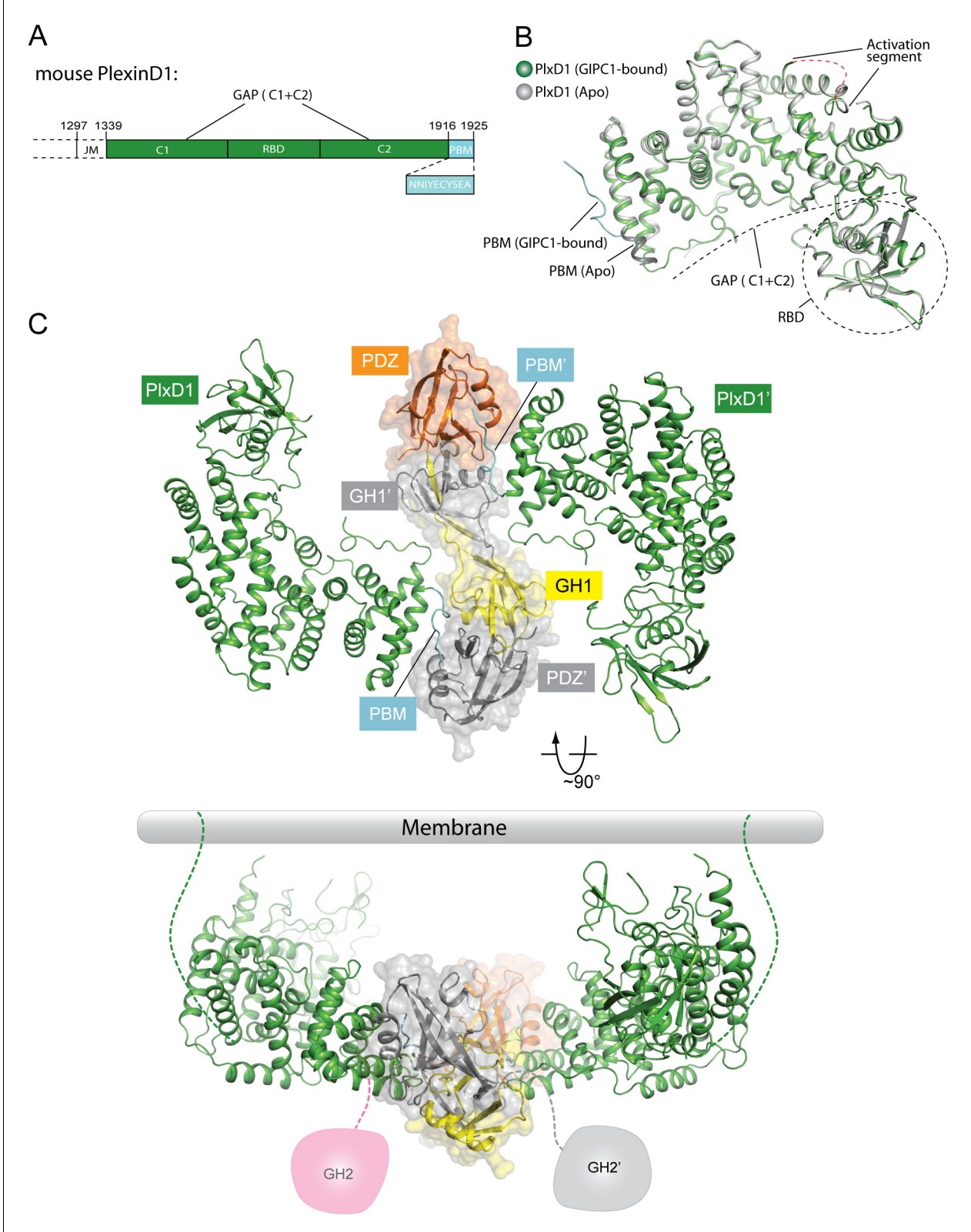

**Figure 3.** Crystal structure of PlexinD1$_{cyto}$ and the PlexinD1$_{cyto}$/GIPC1 complex. (**A**) Domain structure of mouse PlexinD1. The extracellular and transmembrane regions in PlexinD1 are omitted. The crystallization construct includes the regions denoted by the solid boxes. The single-letter amino acid sequence of the PBM is shown. (**B**) Comparison of the structures of apo-PlexinD1$_{cyto}$ and PlexinD1$_{cyto}$ from the PlexinD1$_{cyto}$/GIPC1 complex. (**C**) Two orthogonal views of the PlexinD1$_{cyto}$/GIPC1 complex structure. The dotted lines in green indicate the connection to the membrane by the

*Figure 3 continued on next page*

Figure 3 continued

transmembrane and juxtamembrane (JM) regions of PlexinD1. The linker-GH2 domains in GIPC1, invisible in the structures, are drawn as cartoons. Plx, Plexin.

The following figure supplements are available for figure 3:

**Figure supplement 1.** Electron-density maps for the PBM region in the structures of (**A**) apo-PlexinD1 and (**B**) PlexinD1 in complex with GIPC1.

**Figure supplement 2.** Gel analysis of crystals of the PlexinD1/GIPC1 complex.

**Figure supplement 3.** Comparison of the domain-swapped dimer of the GH1-PDZ domains from apo-GIPC1 and the PlexinD1/GIPC1 complex.

Y1919 in the tail of PlexinD1 plug into a hydrophobic pocket formed at the junction between the GH1 and PDZ domain. Glu1920, Cys1921 and Tyr1922 in the PlexinD1 tail are sandwiched between the PlexinD1 GAP domain and GIPC1, making additional Van der Waals and polar interactions. Most residues involved in these interactions are highly conserved in PlexinD1 from different species but divergent in other plexin family members (*Figure 4B*), supporting the functional relevance of these interactions (*Burk et al., 2017*; *Gay et al., 2011*). These additional interactions likely enhance the affinity and specificity between PlexinD1 and GIPC1. This dual binding mode is analogous to the interaction between class B plexins and PDZ-RhoGEF where the secondary interface significantly increases the binding affinity (*Pascoe et al., 2015*). A superimposition of the PlexinD1$_{cyto}$/GIPC1 complex with apo-GIPC1 shows that the GH2 domain sterically clashes with PlexinD1 (*Figure 4C*), supporting the notion that the GH2 domain in apo-GIPC1 is autoinhibitory and must dissociate before PlexinD1 can bind.

## Structure and oligomerization of the GH2/myosin VI complex

The 'RRL' motif in myosin VI is critical for the interaction with GIPC1 (*Spudich et al., 2007*), but the precise GIPC-binding region was not mapped. We purified various segments of mouse myosin VI containing the 'RRL' motif fused with glutathione sulfur transferase (GST). While longer constructs tended to degrade, the GST-fusion of a conserved 45-residue segment (residues 1052–1096, 'HCBD' in *Figure 5A*; residue numbers based on the short isoform of myosin VI) was stable and interacted with the GH2 domains from all three GIPCs (See below). These observations are consistent with two recent studies showing that this region in myosin VI forms a compact helical domain and is sufficient for GIPC1 binding (*He et al., 2016*; *Wollscheid et al., 2016*). Here, we refer to the 1052–1096 region of myosin VI the helical cargo-binding domain (HCBD) to distinguish it from the C-terminal canonical CBD. We determined the crystal structures of the myosin VI-HCBD in complex with the GH2 domain from GIPC1 and GIPC2, respectively (*Figure 5* and *Figure 5—figure supplement 1* and *Table 1*). The two structures are highly similar (*Figure 5B and C*), and the following discussions will refer to the GIPC1-GH2/myosin VI complex unless otherwise stated.

The HCBD of myosin VI adopts the same two-helix conformation as seen in the recent NMR structure (PDB ID: 2N10), with the 'RRL' motif located at the center of the second helix (*Figure 5B*) (*He et al., 2016*; *Wollscheid et al., 2016*). The GH2 domain of GIPC1 in complex with the myosin VI-HCBD is similar to that in the apo-GIPC1 structure. There are five copies of both myosin VI-HCBD and GIPC1-GH2 molecules in each asymmetric unit, arranged in an alternating pattern (*Figure 5B*). The group of five HCBD and GH2 domains repeats itself in the neighboring unit cell, yielding a continuous array of alternating HCBD and GH2 that extends throughout the crystal (*Figure 5—figure supplement 2A*). Considering one GH2/HCBD pair as a unit, the neighboring units in the array are related roughly by a five-fold right-handed rotational symmetry and a ~22 Å translation. The GIPC2-GH2/myosin VI structure forms the same alternating array, despite being crystallized in a different condition and space group (*Figure 5C* and *Figure 5—figure supplement 2B*). The recurrence of this repeating array argues strongly that this arrangement captures the oligomeric interaction mode between GIPCs and myosin VI that is relevant to function, rather than being a crystal packing artifact. The alternative arrays mediated by the bivalent interactions seen here suggest an unanticipated mechanism for high-order oligomerization of the GIPC/myosin VI complex. The oligomer may collect multiple myosin VI molecules to enhance the processive transport of endocytic vesicles on actin

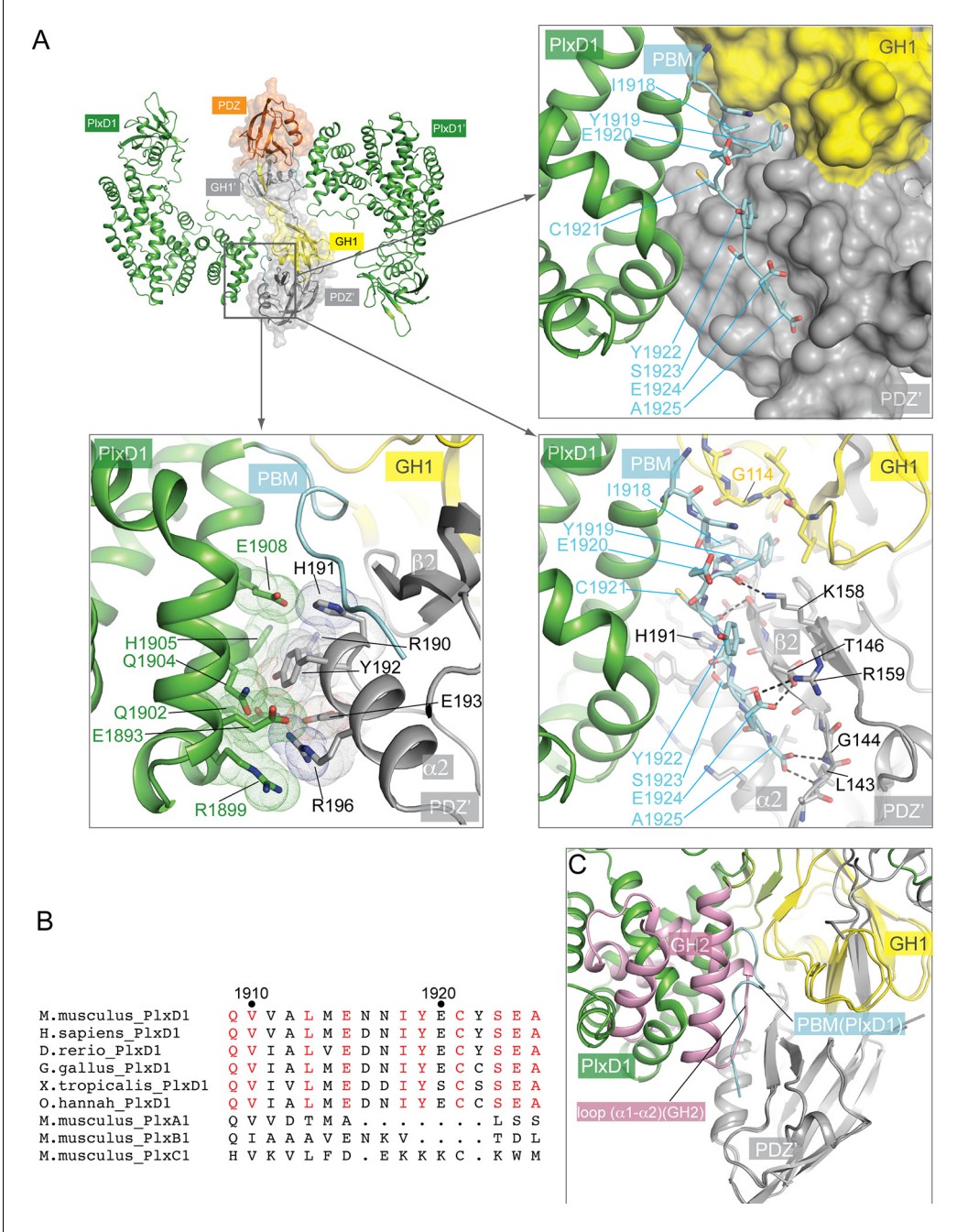

**Figure 4.** Interactions between PlexinD1 and GIPC1. (**A**) Detailed views of the interfaces. The top right panel highlights the packing interactions between the PlexinD1-PBM and GIPC1. The I1918/Y1919 motif in the PMB is accommodated by a hydrophobic pocket between the GH1 and PDZ domains in GIPC1. The lower right panel shows detailed interactions between the PlexinD1-PBM and GIPC1. The lower left panel shows the additional interface between the PlexinD1-GAP domain and the GIPC1-PDZ domain. (**B**) Sequence alignment of the C-terminal region of PlexinD1 from different species and other plexin family members. Residue numbers are based on mouse PlexinD1. (**C**) Superimposition of the structures of the PlexinD1$_{cyto}$/GIPC1 complex and apo-GIPC1 based on the PDZ domain. It is evident that the GIPC1-GH2 domain in the apo-GIPC1 structure clashes with PlexinD1 bound to GIPC1. Plx, Plexin.

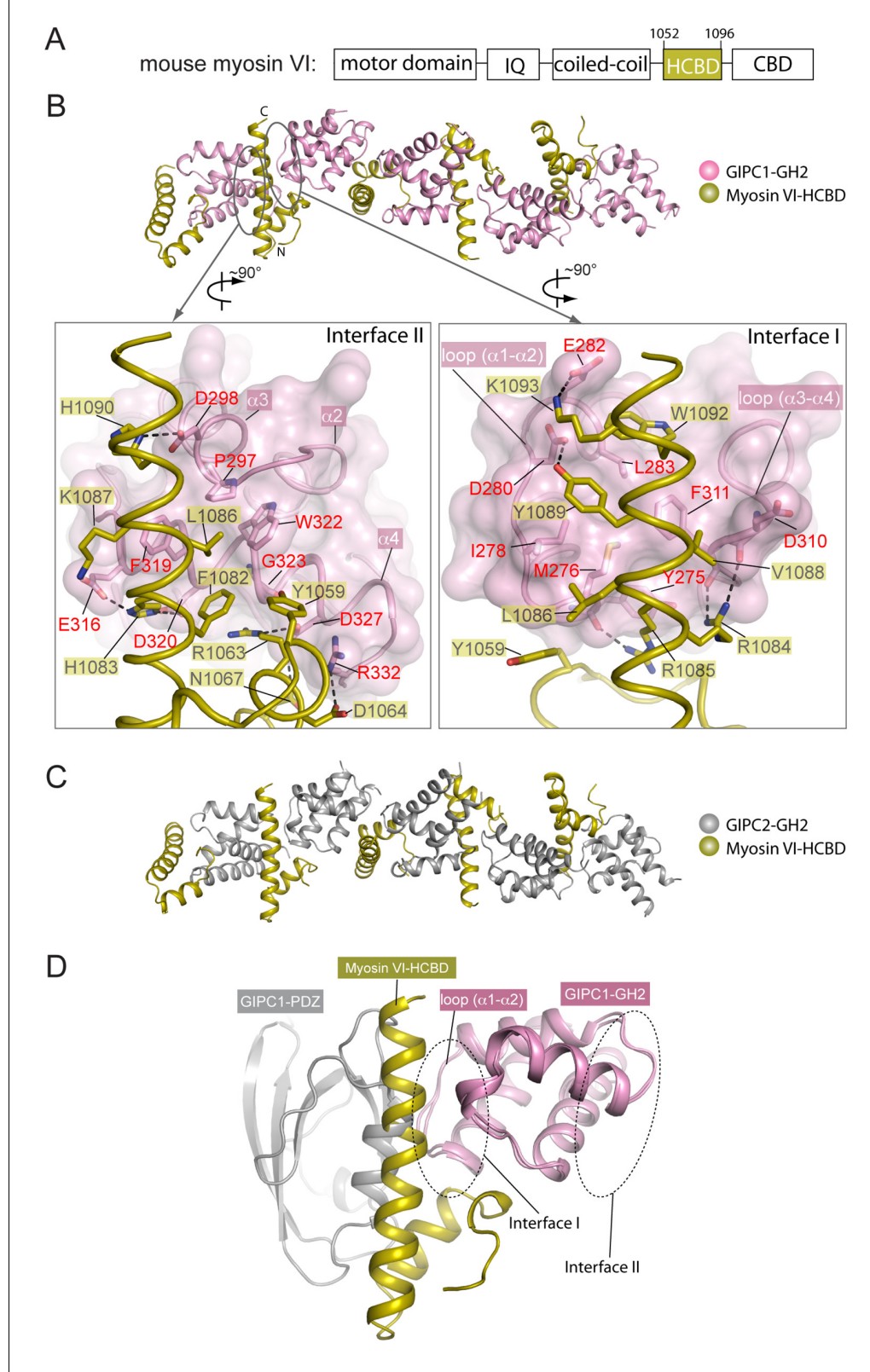

**Figure 5.** Crystal structures of the GH2 domains from GIPC1 and GIPC2 in complex with the myosin VI-HCBD. (**A**) Domain structure of mouse myosin VI. The HCBD is drawn larger than its proportion. (**B**) Structure of the GIPC1-GH2/myosin VI-HCBD complex. The five GH2/HCBD units in the asymmetric unit are shown. The N- and C-termini of one HCBD are labeled. Interfaces I and II between the GIPC1-GH2 and myosin VI-HCBD domains are displayed

*Figure 5 continued on next page*

*Figure 5 continued*

in detail in the two expanded views, respectively. (**C**) Overall structure of the GIPC2-GH2/myosin VI-HCBD complex. The five GH2/HCBD units in the asymmetric unit are shown. (**D**) Superimposition of one GH2/HCBD unit to the apo-GIPC1 structure. The superimposition is based on the GH2 domain. For clarity, only one GH2 and PDZ from the apo-GIPC1 structure are shown. It is evident that the PDZ domain in the apo-GIPC1 structure clashes with the myosin VI-HCBD bound to interface I of the GH2 domain.

The following figure supplements are available for figure 5:

**Figure supplement 1.** Electron-density maps for the HCBD in the structures of (**A**) the GIPC1-GH2/myosin VI-HCBD complex and (**B**) the GIPC2-GH2/myosin VI-HCBD complex.

**Figure supplement 2.** Crystal packing of (**A**) the GIPC1-GH2/myosin VI-HCBD complex and (**B**) the GIPC2-GH2/myosin VI-HCBD complex.

**Figure supplement 3.** Comparison of interfaces I and II in the structures of the GIPC1-GH2/myosin VI-HCBD and GIPC2-GH2/myosin VI-HCBD complexes.

**Figure supplement 4.** Interference of the GIPC-GH2/myosin VI-HCBD interaction by the insert helix in the longer version of myosin VI.

---

filaments, as suggested by the observation that multiple myosin VI dimers attached to the same cargo can coordinate their movements to drive long-distance runs over interlaced actin networks (*Sivaramakrishnan and Spudich, 2009*). Previous studies have suggested that GIPC1 and myosin VI contribute to clustering of cell surface receptors into clathrin-coated pits at the initial stage of endocytosis in certain cell types, particularly those with microvilli (*Hasson, 2003*). In such cells, high-order oligomerization of GIPC1 and myosin VI may make this process more efficient. The anchoring function of the GIPC3/myosin VI complex for stereocilia in the inner ear may also benefit from the enhanced mechanical strength provided by the high-order oligomerization (*Sweeney and Houdusse, 2007*).

Due to the fivefold rotational symmetry, GH2 molecules in the GH2/HCBD oligomer assume five different orientations (*Figure 5B and C*). Considering both the structures of the PlexinD1$_{cyto}$/GIPC1 and GH2/HCBD complexes in the context of the lipid membrane, the ~40 residue flexible PDZ-GH2 linker in GIPCs can readily bridge the distance between the PlexinD1-bound PDZ domain and the GH2 domain in the GH2/HCBD oligomer regardless of the relative orientation of the GH2 domain. Likewise, the predicted flexible regions surrounding the HCBD in myosin VI allow differently oriented HCBDs in the oligomer to connect to the motor domains walking on one or multiple actin filaments. Oligomerization with combinations of translational and rotational symmetries is also seen in the kinase domains of the EGF receptor and the endoplasmic reticulum (ER) stress sensor IRE1, which are transmembrane proteins located on the plasma and ER membrane, respectively (*Huang et al., 2016*; *Korennykh et al., 2009*). In these cases, the flexibility in various linkers has also been proposed to allow differently oriented kinase molecules to converge at the membrane surface and connect to the transmembrane region of the proteins.

## Interactions in the two interfaces between the GIPC-GH2 and myosin VI-HCBD domains

The bivalent interactions between the GIPC-GH2 and myosin VI-HCBD are mediated by two distinct interfaces (*Figure 5B* and *Figure 5—figure supplement 3*). Interface I involves the two loops in the GIPC-GH2 domain, one between helices α1 and α2 and the other between helices α3 and α4 (*Figure 5B*). These two loops interact with Arg1084 and Arg1085 in the 'RRL' motif on one side of the second helix in the myosin VI-HCBD. In addition, hydrophobic residues such as Val1088, Tyr1089 and Trp1092 on the same side of the second helix in the myosin VI-HCBD dock on the hydrophobic patch formed by Tyr275, Met276, Ile278, Leu283, Phe311, etc. in the GIPC1-GH2 domain. Interface I is further expanded by the first helix and the N-terminal extension in the myosin VI-HCBD. Interface II is formed on the opposite side of the HCBD with helix α4 and the N-terminal end of helix α3 in

the GIPC1-GH2 domain (*Figure 5B*). Tyr1059 and Leu1086 (the 'L' in the 'RRL' motif) in the myosin VI-HCBD are sandwiched between interfaces I and II. Phe1082 and Leu1086 from the myosin VI-HCBD and Pro297, Phe319, Trp322 and Gly323 from the GIPC1-GH2 domain together form the hydrophobic core of interface II. There are also several charge- and hydrogen bond-mediated interactions. Interfaces I and II bury ~1300 and ~1100 Å$^2$ solvent accessible surface area, respectively. The two interfaces in the GIPC2-GH2/HCBD complex structure are highly similar to those in the GIPC1-GH2/HCBD complex (*Figure 5—figure supplement 3*).

As mentioned above, the loop between helices α1 and α2 in the GH2 domain blocks the PBM-binding site on the PDZ domain in the apo-GIPC1 structure (*Figure 2B*). Conversely, the inter-domain interaction prevents this loop from forming interface I with HCBD in myosin VI (*Figure 5D*). Therefore, the apo-GIPC1 structure represents the state in which the PDZ and GH2 domains mutually inhibit each other. Interface II in the GH2 domain in the autoinhibited structure of GIPC1 is open for interacting with myosin VI. It is unclear at present whether and how the interface-II-mediated interaction is regulated.

The HCBD region in myosin VI also binds several other adaptor proteins, including optineurin, NDP52 and T6BP, and K63-linked ubiquitin chains (*He et al., 2016*; *Tumbarello et al., 2013*). Alternative splicing generates several isoforms of myosin VI. The longer isoform contains an insert N-terminal to the HCBD, which switches the binding specificity of myosin VI from adaptors such as GIPC1 to clathrin (*Wollscheid et al., 2016*). In the NMR structure of the myosin VI longer isoform (PDB ID: 2N12), a part of the insert forms a helix and packs against the HCBD. A comparison with our GIPC1-GH2/myosin VI-HCBD structure shows that the insert helix interferes with the GH2/HCBD interaction via both interfaces I and II (*Figure 5—figure supplement 4*). These analyses refine our understanding of the structural basis for the different binding specificities between the myosin VI short and long isoforms (*Wollscheid et al., 2016*).

## Mutational analyses of the interactions among PlexinD1, GIPCs and myosin VI

We carried out pull-down experiments using GST-fusion of the myosin VI-HCBD to examine the interactions among PlexinD1, GIPC1 and myosin VI (*Figure 6*). As expected, GST-HCBD was able to pull-down GIPC1 but not PlexinD1$_{cyto}$ (*Figure 6A*). The pull-down of GIPC1 by GST-HCBD is probably primarily mediated by interface II, which is not blocked in the autoinhibited conformation of GIPC1. Mixing the three proteins allowed GST-HCBD to pull-down both GIPC1 and PlexinD1$_{cyto}$. Notably, the presence of Plexin$_{cyto}$ increased the amount of GIPC1 pulled-down substantially. These results are consistent with previous observations and our structure model in which cargo binding releases the autoinhibition in GIPC1, leading to enhanced interaction with the HCBD of myosin VI (*Naccache et al., 2006*). We also conducted the same experiments with GIPC2 and GIPC3. The results were similar, except that the enhancement of GIPC2 binding to GST-HCBD caused by PlexinD1 is more pronounced. These results support the notion that the autoinhibited structure of GIPC1 represents a conserved regulation mechanism of the GIPC family.

We used the three-protein pull-down experiments to examine the interface between PlexinD1 and GIPC1 (*Figure 6B*). Truncation of the three C-terminal residues in the PBM of PlexinD1 (ΔSEA) virtually abolished the interaction with GIPC1. Mutating Ile1918 and Tyr1919 in PlexinD1 (I1918A/Y1919A), which occupy the hydrophobic pocket between GH1 and PDZ in GIPC1 (*Figure 4A*), reduced the binding to GIPC1 substantially (*Figure 6B*). Conversely, a mutation of Gly114 in GIPC1 (G114Y), located at the edge of the hydrophobic packet that accommodates the Ile1918/Tyr1919 motif in PlexinD1 (*Figure 4A*), also reduced the PlexinD1/GIPC1 interaction (*Figure 6B*). These results confirm that the C-terminal 'SEA' motif in PlexinD1 is essential for the binding to GIPC1, and the additional interface beyond the canonical PBM/PDZ interface also contributes the PlexinD1/GIPC interaction.

## Oligomerization of the GIPC1-GH2/myosin VI-HCBD complex in solution

We examined the solution behavior of GIPC1-GH2 and the myosin VI-HCBD both alone and in complex using sedimentation velocity analytical ultracentrifugation (AUC). The GIPC1-GH2 domain at 30 μM displayed a standardized sedimentation coefficient ($s_{20,w}$-value) of ~1.3 S, corresponding to a

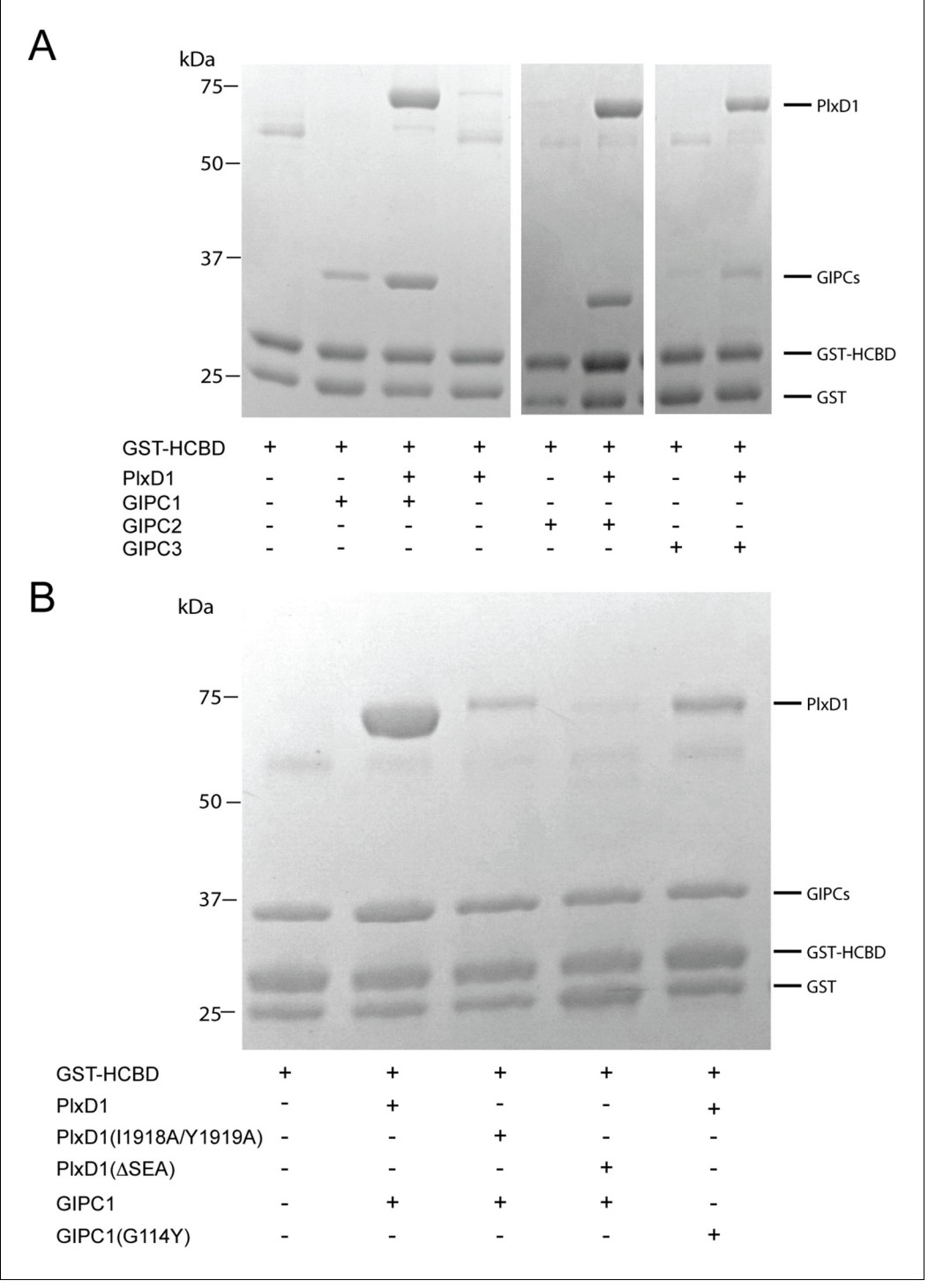

**Figure 6.** Analyses of the interactions of myosin VI-HCBD with GIPCs and PlexinD1. (**A**) Pull-down of GIPC1, GIPC2 and GIPC3 by GST-HCBD in the absence or presence of PlexinD1$_{cyto}$. (**B**) Effects of mutations in PlexinD1 and GIPC1 on the interactions among PlexinD1$_{cyto}$, GIPC1 and myosin VI-HCBD. PlxD1, PlexinD1$_{cyto}$.

monomeric species of ~8 kDa (**Figure 7—figure supplement 1A**). The $s_{20,w}$-value of the myosin VI-HCBD (~5 kDa) at 20 µM was 0.9 S, also consistent with a monomeric species (**Figure 7—figure supplement 1A**). When mixed at a 1:1 molar ratio, the proteins appeared to form multiple complexes (**Figure 7—figure supplement 1A**). We characterized these by examining the weighted-average

sedimentation coefficient ($s_{20,w}^w$) of the entire range of species observed between 1.5 and 6 S. As the overall concentration was increased, the $s_{20,w}^w$ also rose, consistent with the formation of higher order species (*Figure 7A*). When practicable, multi-signal methods (*Balbo et al., 2005*) were used to confirm that both proteins participated in these complexes in an approximate 1:1 molar ratio (*Figure 7— figure supplement 1B and C*). However, we observed no species with $s_{20,w}$-values greater than ~4.8 S (*Figure 7—figure supplement 1A*), and the increase in $s_{20,w}^w$ ceased at concentrations higher than 150 µM (*Figure 7A*). Based on hydrodynamic estimates, 4.8 S could represent a 5:5 complex between the GH2 and HCBD. The observation of an apparent maximum $s_{20,w}^w$-value was unexpected, given the potential of the open-ended GH2/HCBD complex to form infinite oligomers. One possible explanation for this phenomenon is that the formation of very long, linear polymers could be inhibited by the pressures induced by ultracentrifugation (*Harrington and Kegeles, 1973*; *Marcum and Borisy, 1978*).

We designed mutations to test the contributions of both interfaces I and II to the oligomerization. We chose to mutate the GH2 of GIPC1 because most of the residues in the myosin VI-HCBD are located in close proximity to both interfaces I and II, due to the small size of the HCBD. We made a number of GH2 mutants with single mutations in either interface I (Y275A, M276E and I278Q) or II

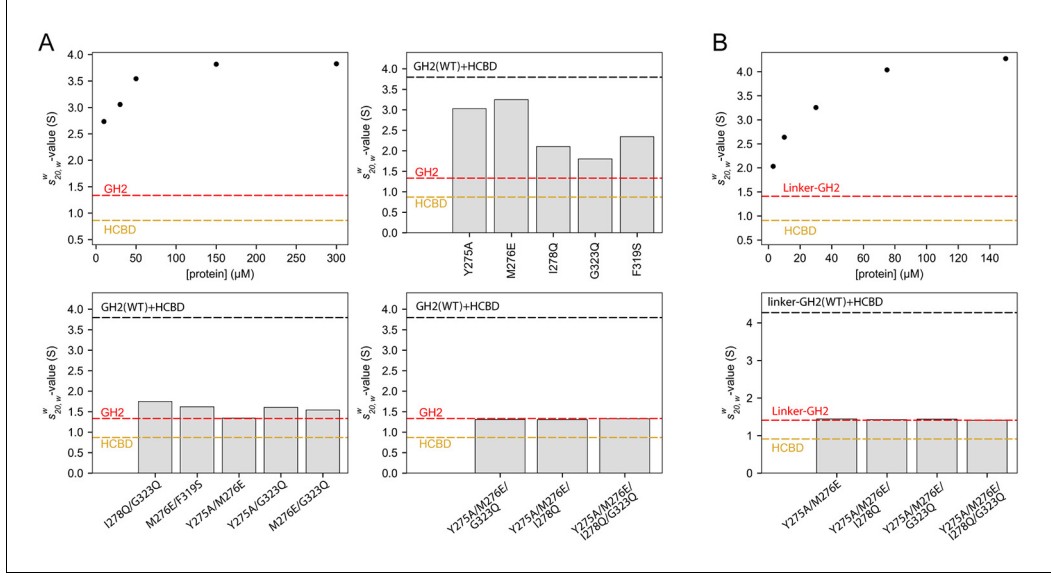

**Figure 7.** Oligomerization of the GIPC1-GH2/myosin VI-HCBD complex in solution. (**A**) Analytical ultracentrifugation (AUC) of the GH2, HCBD and their complexes. The dots in the top-left panel show the $s_{20,w}^w$-values of the wild-type GH2/HCBD complex at concentrations as indicated on the X-axis. Other panels describe the effects of mutating interfacial residues in the GH2 on the $s_{20,w}^w$-value of the GH2/HCBD complex (150 µM). The red and orange dashed lines indicate the $s_{20,w}$-values of the GH2 domain at 30 µM and the HCBD at 20 µM, respectively. The black dashed line indicates the $s_{20,w}^w$-value of the wild-type complex at 150 µM. (**B**) AUC of the linker-GH2 and its complex with the HCBD. The top panel describes the titration analogous to the top left panel of (**A**). The bottom panel shows the effects of interfacial mutations in the linker-GH2 on the $s_{20,w}^w$-value of the linker-GH2/HCBD complex (150 µM). The HCBD at 150 µM sedimented as a major monomeric and minor dimeric species as shown in *Figure 7—figure supplement 1*. The $s_{20,w}$-value of the monomeric species is indicated by the orange dashed lines. The $s_{20,w}$-value of the linker-GH2 (red dashed line) is the same at both 38 µM and 200 µM. The black dashed line indicates the $s_{20,w}^w$-value of the wild-type complex at 150 µM.

The following figure supplements are available for figure 7:

**Figure supplement 1.** AUC data of the HCBD, GH2, linker-GH2 and the complexes.

**Figure supplement 2.** Salt-dependent reversible precipitation of the complex between the linker-GH2 of GIPC1 and myosin VI-HCBD.

(F319S and G323Q). The results showed that these mutations led to decreased $s_{20,w}^w$-values of the GH2/HCBD complex (*Figure 7A*), implying that the mutations disrupted oligomer formation. We combined the single mutations to generate a number of double, triple and quadruple mutants of the GIPC1-GH2 domain. The combined mutations further decreased the $s_{20,w}^w$-values of the complex, and in some cases virtually eliminated complex formation (*Figure 7A*).

We also tested the interaction of the myosin VI-HCBD with a linker-GH2 construct of GIPC1 that contains both the PDZ-GH2 linker and GH2 domain. Surprisingly, mixing the linker-GH2 and HCBD led to heavy precipitation (*Figure 7—figure supplement 2*). The precipitation is unlikely due to denaturation of the proteins, as it could be reversed by increasing the salt concentration from 150 mM to 500 mM. These observations, reminiscent of the reversible precipitation of IRE1 due to the high-order oligomerization (*Korennykh et al., 2009*), suggest that the PDZ-GH2 linker promotes the polymerization of the myosin VI-HCBD/GIPC1-GH2 complex. We analyzed the linker-GH2 and its complex with the HCBD by AUC, with 500 mM salt in the buffer to prevent precipitation. The linker-GH2 protein sedimented as a monomer at both 38 μM and 200 μM, with a $s_{20,w}$-value value of ~1.4 S (*Figure 7—figure supplement 1D and E*). The HCBD at 150 μM under the same condition sedimented predominantly as a monomer ($s_{20,w}$-value of 0.9 S), and a minor dimer peak ($s_{20,w}$-value of 1.6 S) (*Figure 7—figure supplements 1D* and *2F*). Similar to the GH2/HCBD complex, the average sizes of the linker-GH2/HCBD complex increased as a function of concentration, with the largest observed $s_{20,w}$-value of ~5.5 S (*Figure 7B* and *Figure 7—figure supplement 1D*). This value is slightly higher than that observed for the GH2/HCBD mixture at the same concentration, which could be due to the larger mass of the linker-GH2 construct and/or the formation of higher order oligomers. We introduced some double, triple and quadruple mutations as described above to the linker-GH2 construct. These mutations virtually eliminated the complex formation between the linker-GH2 and HCBD (*Figure 7B*).

These results together demonstrate that GIPC1-GH2 and myosin VI-HCBD form high-order oligomers in solution, and both interfaces I and II as seen in our crystal structures are required for the oligomerization. The PDZ-GH2 linker in GIPC1 appears to enhance the oligomerization, but the precise mechanism by which it does so remains unclear at present.

## Oligomerization of Sema3E/PlexinD1, GIPC1 and myosin VI in cells

To analyze the interactions among PlexinD1, GIPC1 and myosin VI in mammalian cells, we generated COS-7 cells stably expressing full-length mouse PlexinD1 (*Figure 8—figure supplement 1*). These cells were then transiently co-transfected with myosin VI and GIPC1. In these cells, a fraction of PlexinD1 was incompletely processed and remained in the ER or Golgi apparatus (*Figure 8—figure supplement 1B*). To selectively image fully processed PlexinD1 molecules, we treated the cells with the Sema3E ligand labeled with Alexa Fluor-555 and used it as a proxy for PlexinD1. Sema3E/PlexinD1, GIPC1 and myosin VI co-localized very well and formed large puncta in cells expressing wild-type PlexinD1 (*Figure 8*). In cells expressing PlexinD1(ΔSEA), GIPC1 and myosin VI also formed puncta together, but Sema3E did not localize to these puncta (*Figure 8*). There is virtually no large puncta with Sema3E/PlexinD1, GIPC1 and myosin VI co-localized in these cells. Replacing wild-type GIPC1 with the G323Q mutant, which disrupts interface II in the GH2 domain for myosin VI, also abolished formation of large puncta of co-localized Sema3E/PlexinD1, GIPC1 and myosin VI (*Figure 8*). These results support the notion that Sema3E, PlexinD1, GIPC1 and myosin VI can form high-order oligomers in cells, which is dependent on both the PlexinD1/GIPC1 interaction and the linear oligomerization of the GIPC1/myosin VI complex.

## Discussion

Ligand-induced activation of cell surface receptors often leads to their endocytosis. The primary function of GIPCs in this process is to tether the receptors on uncoated endocytic vesicles to myosin VI, which transports the vesicles to fuse with early endosomes (*Katoh, 2013*). Our structural analyses reveal how the GIPC proteins are autoinhibited, and how they are activated upon binding of PlexinD1 or other cargos and consequently interact with myosin VI (*Figure 9*). The autoinhibition of both GIPCs and myosin VI (*French et al., 2017*) helps suppress their interaction in the absence of cargo proteins, avoiding futile ATP consumption and walking of myosin VI. The linear array of the GIPC/myosin VI complex formed through their mutual bivalent binding provides an elegant mechanism for

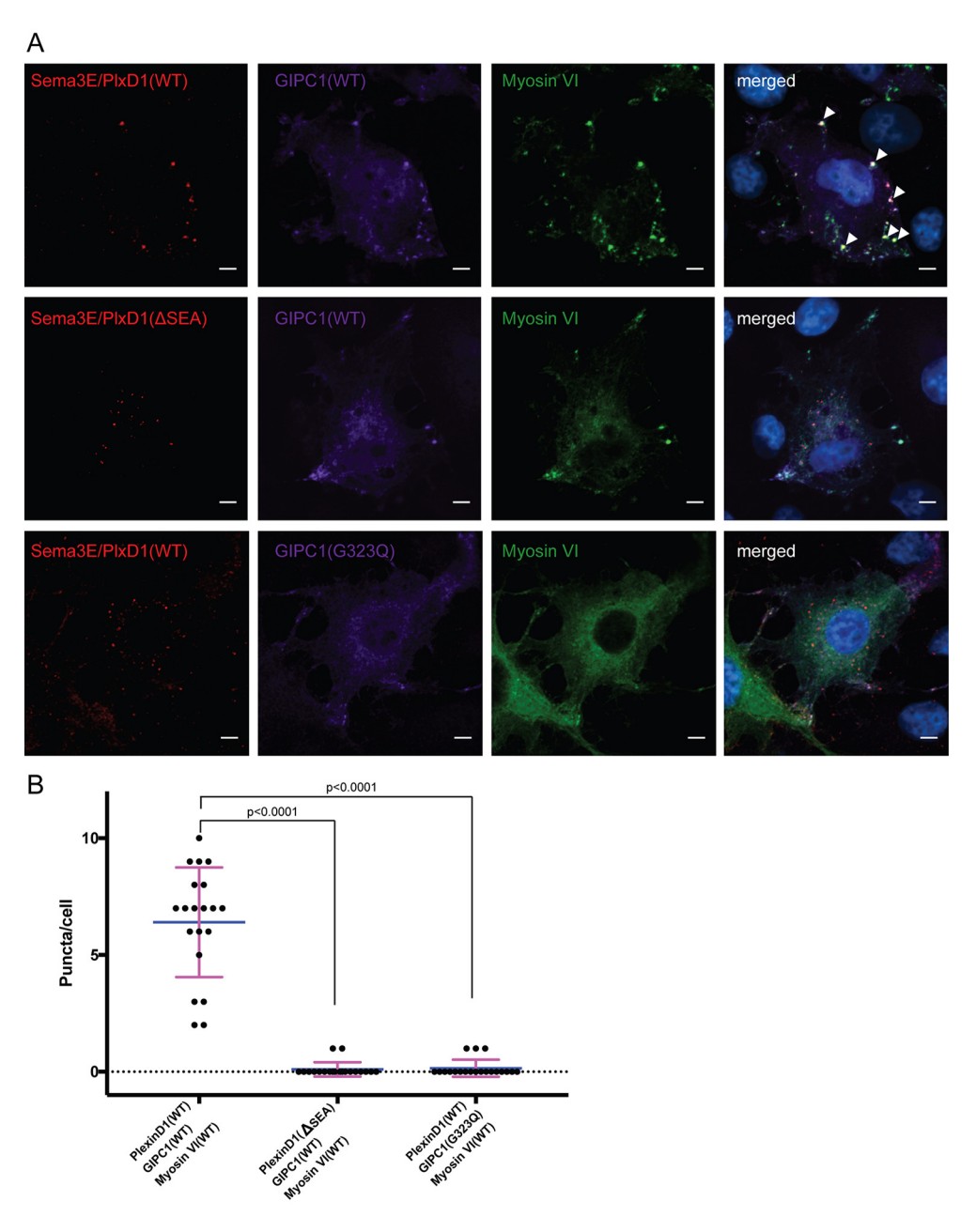

**Figure 8.** Co-localization and oligomerization of Sema3E/PlexinD1 with GIPC1 and myosin VI in mammalian cells. (**A**) Representative fluorescence images of cells. Arrow heads highlight large puncta in which Sema3E/PlexinD1 (red), GIPC1 (purple) and myosin VI (green) co-localize. Nuclei were stained with DAPI (blue). Scale bar, 5 μm. Images shown are representative from three independent samples of each group. (**B**) Quantification of puncta in each group of cells. Puncta larger than 1 μm$^2$ and containing Sema3E/PlexinD1, GIPC1 and myosin VI are counted for 20 cells from each group. Each dot in the plot represents one cell. Mean and standard deviations are shown as the blue and magenta bars, respectively. p-Values are determined by two-tailed Student's t-test.
The following figure supplement is available for figure 8:

**Figure supplement 1.** Protein expression analyses of COS-7 cells stably expressing PlexinD1.

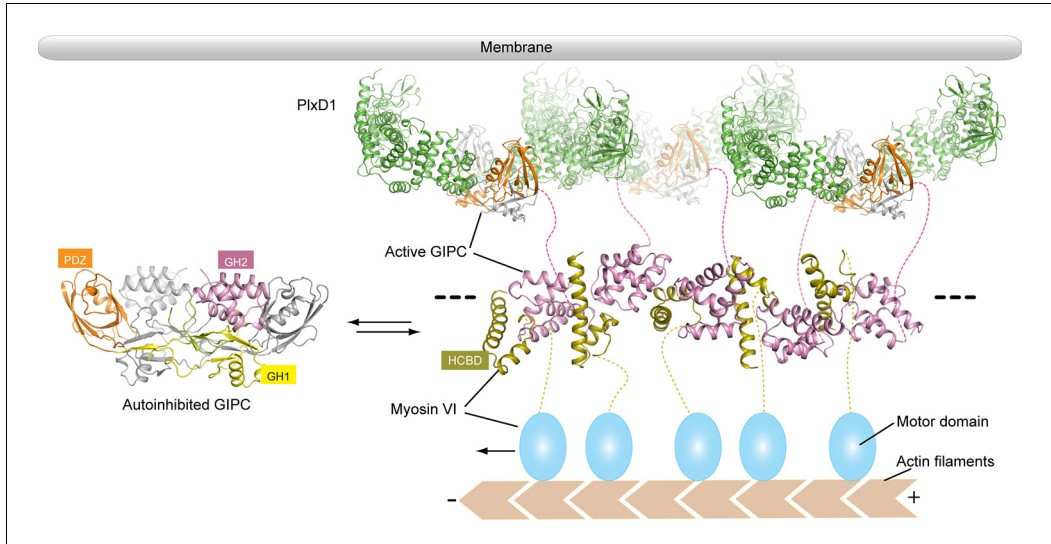

**Figure 9.** Model of the regulated oligomerization of PlexinD1, GIPCs and myosin VI. Unbound GIPCs adopt the autoinhibited conformation (left). Binding of cargo such as PlexinD1 to GIPCs releases the autoinhibition, promoting their interaction with myosin VI and formation of the linear oligomer (right). The black dashed lines indicate that the oligomer can extend further, and multiple such oligomers can assemble into two-dimensional arrays. The transmembrane and extracellular regions of PlexinD1 are omitted for clarity.

myosin VI oligomerization. PlexinD1 and many other GIPC-binding cell surface receptors undergo ligand-induced dimerization, which combined with the dimeric nature of GIPCs can potentially cross-link the GIPC/myosin VI linear oligomer into large two-dimensional assemblies (*Figure 9*). Such assemblies may be more efficient in performing the cellular functions of myosin VI, including clustering cargos to clathrin-coated pits, endocytic vesicle transport and maintenance of stereocilia in the inner ear. There is evidence that myosin V also forms large oligomers for processive transport (*Chung and Takizawa, 2010*; *Krementsova et al., 2011*). Oligomerization of myosin adds another instance to the increasingly prevalent higher order assemblies involved in a variety of cellular processes (*Wu and Fuxreiter, 2016*).

The crystal structures presented here provide a framework for understanding how mutations in PlexinD1, GIPCs and myosin VI lead to diseases. Many germ-line mutations in GIPC3 and myosin VI are causatively associated with hereditary deafness (*Ahmed et al., 2003*; *Ammar-Khodja et al., 2015*; *Avraham et al., 1995*; *Charizopoulou et al., 2011*; *Melchionda et al., 2001*; *Rehman et al., 2011*; *Siddiqi et al., 2014*). Mapping the GIPC3 mutations to the crystal structure of apo-GIPC1 suggest that many of the mutations likely destabilize the GIPC3 protein or disrupt the interaction with myosin VI (*Table 2* and *Figure 10*). Deafness caused by these mutations is therefore likely due to loss of the anchoring function of the GIPC3/myosin VI complex in stereocilia. Dysregulation of receptor tyrosine kinases such as Vascular Endothelial Factor Receptor two and Insulin-like Growth Factor-1 Receptor can lead to cancer. GIPC-mediated endocytosis and intracellular transport are important aspects of signal regulation of these receptors (*Katoh, 2013*). Some somatic mutations in GIPCs found in human cancer (*Katoh, 2013*) also map to structural elements that are important for the protein stability or the interaction with myosin VI (*Table 2* and *Figure 10*).

While other plexin family members are not known to bind GIPCs (*Burk et al., 2017*; *Gay et al., 2011*), some of them form holo-receptors with Neuropilin 1 or 2, which contains a PBM that interacts with GIPCs (*Cai and Reed, 1999*). In *Drosophila*, class A plexin forms a holo-receptor with the receptor guanylyl cyclase Gyc76C, which binds GIPC1 and facilitates plexin-mediated motor axon guidance (*Chak and Kolodkin, 2014*). GIPCs therefore can use diverse mechanisms to regulate plexin signaling. Intriguingly, the PBM of PlexinD1 is in principle available for binding GIPCs constitutively, raising the question whether and how the plexin/GIPC interaction is regulated. One potential mechanism for regulating the interaction in PlexinD1 may be phosphorylation of conserved Tyr1919 and

**Table 2.** Structural mapping of disease-associated mutations in GIPCs. Listed mutations are based on (**Ammar-Khodja et al., 2015**; **Katoh, 2013**). HNSCC, head and neck squamous cell carcinoma. Mutations are mapped to GIPC1 based on the sequence alignment of GIPC1, 2 and 3 as shown in **Figure 1—figure supplement 2**.

| Gene | Mutation | Disease | Corresponding residue in GIPC1 | Location and function of the mutated residue | Possible effect of the mutation |
|---|---|---|---|---|---|
| GIPC1 | F319L | HNSCC | F319 | GH2, Interface II for myosin VI | Weaken binding to myosin VI |
| GIPC2 | F74Y | Colorectal cancer | F90 | GH1, Hydrophobic core | Destabilize the structure |
| GIPC2 | G102E | Ovarian cancer | G118 | GH1, Domain-swapped dimer Interface | Destabilize the structure |
| GIPC3 | M88I | Familial hearing loss | M109 | GH1, Inter-domain interface with PDZ | Destabilize the structure |
| GIPC3 | G94D | Familial hearing loss | G115 | GH1, autoinhibitory interface with GH2, Part of the hydrophobic pocket accommodating the I1918/Y1919 motif in PlexinD1 | Disrupt the autoinhibited state, Weaken cargo binding |
| GIPC3 | H170N | Familial hearing loss | H191 | PDZ, Key PBM binding residue | Weaken cargo binding |
| GIPC3 | T221I | Familial hearing loss | T242 | PDZ/GH2 Linker, Autoinhibitory interface with GH2 | Alter the autoinhibited state |
| GIPC3 | M255K | Familial hearing loss | M276 | GH2, Autoinhibitory interface with PDZ, Interface I for myosin VI | Alter the autoinhibited state, weaken binding to myosin VI |
| GIPC3 | G256D | Familial hearing loss | G277 | GH2, Autoinhibitory interface with PDZ Interface I for myosin VI | Alter the autoinhibited state, weaken binding to myosin VI |
| GIPC3 | L262R | Familial hearing loss | L283 | GH2, Autoinhibitory interface with PDZ, Interface I for myosin VI | Alter the autoinhibited state, weaken binding to myosin VI |

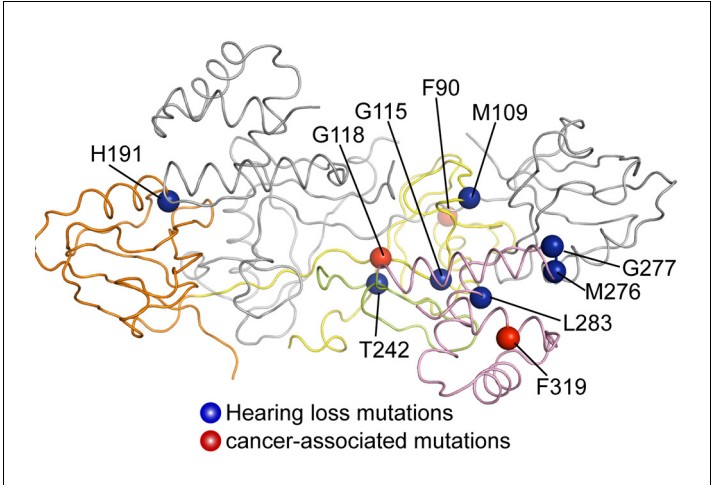

**Figure 10.** Mapping disease-associated mutations in GIPCs to the GIPC1 structure. The structure is shown in the same view and color scheme as in the right panel of **Figure 1B**. The mutations listed in **Table 2** are mapped to the colored subunit in the dimer.

Ser1923, which would likely prevent GIPC binding based on the PlexinD1/GIPC1 complex structure. Interestingly, the sequence context of Tyr1919 resembles the consensus phosphorylation site for the Src family kinases (*Gay et al., 2011*). Future studies will determine the potential role of PlexinD1 phosphorylation on the formation of the PlexinD1/GIPC1 complex.

## Materials and methods

### Protein expression and purification

The coding regions for mouse PlexinD1$_{cyto}$ with (residues 1297–1925) and without the JM-segment (residues 1339–1925), mouse GIPC1 GH1-PDZ-GH2 version 1 (residues 52–333), GIPC1 GH1-PDZ-GH2 version 2 (residues 52–327), GIPC1 linker-GH2 (residues 217–333), GIPC1 GH2 (residues 258–333), mouse GIPC2 GH1-PDZ-GH2 (residues 40–314), GIPC2 GH2 (residues 238–314) and mouse GIPC3 GH1-PDZ-GH2 (residues 20–307) were sub-cloned into a modified pET-28a vector (Novagen). Mutations were introduced by polymerase chain reaction-based mutagenesis. Proteins expressed using these constructs contain an N-terminal His$_6$-tag and human rhinovirus 3C protease cleavage site. Proteins were overexpressed in the bacterial strain BL21(DE3) and purified by Ni$^{2+}$ ion affinity chromatography. The His$_6$-tag was removed by treatment with the 3C protease. Proteins were further purified using ion-exchange and gel filtration chromatography. The GH1-PDZ-GH2 domains of GIPC1 showed two peaks on gel filtration chromatography, presumably corresponding to the dimer and monomer, respectively. The monomeric proteins were unstable and precipitated when concentrated. The dimeric proteins were collected and concentrated for subsequent experiments. GIPC2 and GIPC3 both ran as a single peak corresponding to the dimer species. The coding region for the HCBD of mouse myosin VI (residues 1052–1096; residues numbers based on the short version myosin VI, GeneBank number BC144917) was inserted into the pGEX-6p-1 vector (GE healthcare, Pittsburgh, PA). The GST-HCBD fusion protein was overexpressed in BL21(DE3) and purified using glutathione affinity chromatography followed by gel filtration. To produce the HCBD of myosin VI without the GST moiety, the fusion protein was treated with the 3C protease and GST was removed by passing through a glutathione affinity column. All the proteins were concentrated using Amicon concentrators (Millipore, Billerica, MA) and stored at −80°C before use.

The coding region for mouse Sema3E (residues 1–678) with a C-terminal His$_8$-tag was cloned into the pEZT-BM vector and expressed as a secreted protein in HEK293S-GnTI$^-$ cells (ATCC, catalogue #CRL-3022; not independently authenticated) by using the BacMam system (*Morales-Perez et al., 2016*). The protein was captured by using Ni$^{2+}$-NTA beads from the culture medium and further purified by gel filtration chromatography. Sema3E was labeled with Alexa Fluor-555 (Invitrogen, Waltham, MA) according to the manufacture's instruction. Briefly, Sema3E in phosphate buffered saline (PBS) at 10 mg/ml was incubated with amine-reactive Alexa Fluor-555 for 1 hr at room temperature. Sema3E conjugated with Alexa Fluor-555 was purified by gel filtration chromatography, concentrated and stored at −80°C. Alkaline phosphatase-tagged Sema3E (AP-Sema3E) was expressed in transiently transfected HEK293T cells (ATCC, Manassas, VA, catalogue #CRL-3216; not independently authenticated) as described previously (*He et al., 2009*).

### Crystallization and structure determination

Initial crystallization trials were carried out in 96-well plates using the sitting-drop vapor diffusion method at 20°C. Crystallization conditions were optimized in 24-well plates using the hanging-drop vapor diffusion method. For crystallizing the PlexinD1$_{cyto}$/GIPC1, GIPC1-GH2/myosin VI-HCBD and GIPC2-GH2/myosin VI-HCBD complexes, the interacting proteins were mixed at an equimolar ratio and subjected to crystallization screens directly. The optimized crystallization condition for PlexinD1 (residues 1333–1925, 10 mg/ml) included 0.1 M imidazole (pH 7.8) and 15% (w/v) PEG8000. The condition for GIPC1 (residues 52–327, 5 mg/ml) included 0.2 M Li$_2$SO$_4$, 0.1 M HEPES pH 7.5 and 25% (w/v) PEG3350. The condition for the complex (5 mg/ml) between PlexinD1 and GIPC1 (residues 52–333) was 0.1 M Bis-Tris pH 5.5 and 2 M (NH$_4$)$_2$SO$_4$. The condition for the GIPC1-GH2(258–333)/myosin VI-HCBD(1052–1094) complex (15 mg/ml) was 0.1 M Bis-Tris pH 6.0 and 2 M (NH$_4$)$_2$SO$_4$. The condition for the GIPC2-GH2(238–314)/myosin VI-HCBD(1052–1094) complex (15 mg/ml) was 0.1 M MES pH 6.5% and 10% (w/v) PEG20000. Crystals were cryo-protected in the crystallization buffers supplemented with 20% (v/v) ethylene glycol or glycerol, and flash-cooled in liquid nitrogen before

data collection. Diffraction data were collected at 100 K on beamline 19ID at the Advanced Photon Sources (Argonne National Laboratory, Argonne, IL). The data were processed by using the HKL2000 software package (*Otwinowski and Minor, 1997*).

All the structures were solved by molecular replacement (MR) using Phaser in the Phenix software package (*Adams et al., 2010*; *McCoy et al., 2007*). The structure of PlexinD1$_{cyto}$ was solved using the PlexinA3$_{cyto}$ structure (PDB ID: 3IG3) with the JM-segment removed as the search model. The PlexinD1$_{cyto}$ structure was then used as the search model to obtain the initial phases for the PlexinD1$_{cyto}$/GIPC1 complex structure, which generated an electron-density map that showed clear features of the PDZ domain. The PDZ domain of GIPC2 (PDB ID: 3GGE) was manually placed into the density map using the program Coot (*Emsley and Cowtan, 2004*). Subsequent iterative model building and refinement using Phenix and Coot, respectively, allowed the GH1 domain of GIPC1 to be built and refined. While the GH2 domain was present in the protein used for crystallization, it was completely disordered and invisible in the electron-density map.

The individual GH1 and PDZ domains from the PlexinD1$_{cyto}$/GIPC1 complex structure were used as the search models to solve the structure of apo-GIPC1. The initial electron-density map showed weak features of the GH2 domain, which were improved through iterative manual model building in Coot, automatic building using the Autobuild module and refinement with the Refine module in Phenix. Despite the high resolution (1.9 Å), the R$_{free}$ remained above 30% after many rounds of refinement. Analyses of the diffraction data with the Xtriage module in Phenix suggested that the diffraction data were twinned with the twin operator of (h, -k, -l). Refinement with the twin operator applied in Phenix led to the final model with R$_{free}$of ~ 17.5%. The refined twin fraction is ~33%. The GH2 domain from the refined apo-GIPC1 structure was then used as the search model to solve the structures of HCBD of myosin VI in complex with GIPC1-GH2 and GIPC2-GH2, respectively. For both structures, the electron-density maps calculated from the five protomers of GH2 in the asymmetric units were clear enough to allow the HCBD of myosin VI to be modeled via manual building in Coot and Autobuild in Phenix.

Data collection and refinement statistics are listed in the *Table 1*. Model quality was checked by using MolProbity (*Chen et al., 2010*). The structure figures were generated in Pymol (http://www.pymol.org/).

## Pull-down assays with purified proteins

The GST fusion protein of HCBD of myosin VI was incubated with GIPC1, GIPC2 or GIPC3 in the absence or presence of PlexinD1$_{cyto}$(with JM) in the pull-down buffer containing 10 mM Tris (pH 8.0), 150 mM NaCl and 1 mM DTT at room temperature for 30 min before the pull-down experiments. The concentrations of the individual proteins in the initial 20 µl mix were all at 1 µM. The same conditions were used for incubating the mutants. To perform the pull-down experiments, 20 µl washed glutathione beads were added and incubated for an additional 30 min at room temperature. Beads were pelleted and washed with 1 ml of the pull-down buffer three times. Proteins remaining on the beads were extracted with standard SDS-PAGE loading buffer containing 1% (w/v) SDS, resolved in SDS-PAGE and stained with Coomassie blue R250.

## Analytical ultracentrifugation

For ultracentrifugation experiments of the GH2-C-terminal protein of GIPC1 (residues 258–333) and its complex with the HCBD of myosin VI, the centrifugation buffer contained 10 mM Tris (pH 8.0), 150 mM NaCl and 1 mM TCEP. The complex between the linker-GH2-C-terminal protein and HCBD precipitated heavily in this buffer. Therefore, for the ultracentrifugation experiments with linker-GH2-C-terminal, the concentration of NaCl in the buffer was increased to 500 mM. The individual proteins and protein complexes (at 1:1 molar ratio) at various concentrations were prepared and incubated overnight at 20°C. The samples (~400 µl) were loaded into the 'sample' side of dual-sectored charcoal-filled Epon centerpieces, while the 'reference' sectors were loaded with the centrifugation buffer of the same volume. Filled cells were loaded into a An50Ti rotor and equilibrated for 2 hr under vacuum in a Beckman Coulter Optima XL-I ultracentrifuge at 4°C prior to centrifugation. Data were acquired at 50,000 rpm at 4°C via absorbance at 305 nm and interference optics. Data were analyzed using the *c(s)* methodology in the program SEDFIT (*Schuck, 2000*). Multi-signal

analyses were carried out in SEDPHAT (*Balbo et al., 2005*). All figures featuring *c(s)* distributions were rendered with the program GUSSI (*Brautigam, 2015*).

## Fluorescence imaging of cells

The cDNAs of full-length mouse PlexinD1(WT) and PlexinD1(ΔSEA) were inserted into the Lentiviral vector pTY with the Elongation Factor 1a (EF1a) promoter (Clontech, Mountain View, CA). The constructs were transfected into HEK293T cells to generate Lentiviruses, which were used to infect COS-7 cells (ATCC, catalogue #CRL-1651). Cells were not independently authenticated, but checked by 4,6-Diamidino-2-phenylindole (DAPI) staining to ensure no microplasma contamination. Cells were cultured in DMEM medium (Invitrogen) supplemented with 1% (V/V) penicillin/streptomycin and 10% (V/V) FBS with 5% $CO_2$ at 37°C. Blasticidin (10 μg/ml) was used to select for infected COS-7 cells stably expressing either PlexinD1(WT) or PlexinD1(ΔSEA). Cell surface expression of the PlexinD1 proteins was confirmed by staining with AP-Sema3E (2 nM) as described previously (*He et al., 2009*), except that Sema3E treatment was conducted on ice for 15 min to prevent cell collapse and endocytosis. Whole cell distribution of PlexinD1 was detected by immune-staining with an anti-PlexinD1 antibody (Abcam, San Francisco, CA, catalogue #ab96313) and an Alexa-555-labeled goat anti-rabbit secondary antibody (Invitrogen, catalogue #A-21428). Cells were imaged with a 40X objective and a CCD camera mounted on a Nikon A1R confocal microscope. The excitation and emission wavelengths were 555 nm and 572 nm, respectively.

To analyze the co-localization of Plexin D1 with GIPC1 and myosin VI, human myosin VI short version (GeneBank number U90236.2) was inserted into pEGFP-C3 (Clontech) for expression of N-terminal GFP-fused myosin VI. GIPC1 wild-type and the G323Q mutant were cloned into pcDNA3.1 (Invitrogen) with a C-terminal HA-tag. The myosin VI plasmid was co-transfected with the GIPC1 wild-type or the G323Q mutant plasmid with Fugene 6 (Promega, Fitchburg, WI) into the PlexinD1-expressing COS-7 cells described above. Twenty-four hours after transfection, cells were treated with 15 nM Alexa Fluor-555-labelled Sema3E for 12 min at 37°C. Cells were permeabilized with 0.05% (w/v) saponin for 10 min and non-specific binding sites were blocked by Quenching buffer (0.01% (w/v) saponin, 2% (w/v) BSA, 0.1% (w/v) lysine in PBS, pH7.4) for 30 min. HA-tagged GIPC1 was stained with a chicken anti-HA antibody (Novus, Littleton, CO, catalogue #NB600-361) and an Alexa Fluor 647-labelled goat anti-chicken antibody (Invitrogen, catalogue #1806124). Cells were imaged with a 60 × 1.4 numerical aperture objective and a CCD camera mounted on a Nikon A1R confocal microscope. Z-stacks were obtained and collapsed into 2D images. Excitation wavelengths used were 405 nm (for DAPI), 488 nm (for GFP-myosin VI), 555 nm (for Alexa Fluor-555 labeled Sema3E) and 647 nm (for GIPC1 stained with Alexa Fluor-647-labeled antibody). The corresponding emission wavelengths were 421 nm, 519 nm, 572 nm and 665 nm, respectively.

## Acknowledgements

We thank Ryan Hibbs, Collen Noviello, Hongtao Yu and Conggang Zhang for technical help, Marcel Mattlen and Sandra Schmid for discussions, the Structural Biology Laboratory at the University of Texas Southwestern Medical Center (UTSW) and the staff of beamline 19ID at Advanced Photon Source (APS) for assistance in X-ray data collection. XZ is a Virginia Murchison Linthicum Scholar in Medical Research at UTSW. The work is supported in part by National Institutes of Health grants GM088197 (to XZ) and R01HL133687 (to JT-V); Welch Foundation Grant I-1702 (to XZ). Results shown in this report are derived from work performed at the Argonne National Laboratory, Structural Biology Center at APS. Argonne is operated by UChicago Argonne, LLC, for the U.S. Department of Energy, Office of Biological and Environmental Research under Contract DE-AC02-06CH11357.

## Additional information

### Funding

| Funder | Grant reference number | Author |
| --- | --- | --- |
| National Institutes of Health | GM088197 | Guijun Shang<br>Rui Chen |

|  |  | Xuewu Zhang |
| --- | --- | --- |
| Welch Foundation | I-1702 | Guijun Shang<br>Rui Chen<br>Xuewu Zhang |
| National Institutes of Health | R01HL133687 | Jesús Torres-Vázquez |

The funders had no role in study design, data collection and interpretation, or the decision to submit the work for publication.

## Author contributions

GS, Conceptualization, Data curation, Formal analysis, Validation, Investigation, Visualization, Methodology, Writing—original draft, Writing—review and editing; CAB, Data curation, Formal analysis, Supervision, Validation, Investigation, Visualization, Methodology, Writing—original draft, Writing—review and editing; RC, Validation, Investigation, Writing—review and editing; DL, Investigation, Writing—review and editing; JT-V, Conceptualization, Writing—review and editing; XZ, Conceptualization, Resources, Data curation, Formal analysis, Supervision, Funding acquisition, Validation, Investigation, Visualization, Methodology, Writing—original draft, Project administration, Writing—review and editing

## Author ORCIDs

Xuewu Zhang, http://orcid.org/0000-0002-3634-6711

# Additional files

## Major datasets

The following datasets were generated:

| Author(s) | Year | Dataset title | Dataset URL | Database, license, and accessibility information |
| --- | --- | --- | --- | --- |
| Shang G, Zhang X | 2017 | Structure of apo-PlexinD1 | http://www.rcsb.org/pdb/explore/explore.do?structureId=5V6R | Publicly available at the RCSB Protein Data Bank (accession no: 5V6R) |
| Shang G, Zhang X | 2017 | Structure of apo-GIPC1 | http://www.rcsb.org/pdb/explore/explore.do?structureId=5V6B | Publicly available at the RCSB Protein Data Bank (accession no: 5V6B) |
| Shang G, Zhang X | 2017 | Structure of the PlexinD1/GIPC1 complex | http://www.rcsb.org/pdb/explore/explore.do?structureId=5V6T | Publicly available at the RCSB Protein Data Bank (accession no: 5V6T) |
| Shang G, Zhang X | 2017 | Structure of the GIPC1-GH2/Myosin VI-HCBD complex | http://www.rcsb.org/pdb/explore/explore.do?structureId=5V6E | Publicly available at the RCSB Protein Data Bank (accession no: 5V6E) |
| Shang G, Zhang X | 2017 | Structure of the GIPC2-GH2/Myosin VI-HCBD complex | http://www.rcsb.org/pdb/explore/explore.do?structureId=5V6H | Publicly available at the RCSB Protein Data Bank (accession no: 5V6H) |

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
