## [Decision Letter]

Thank you for submitting your article "Structure analyses reveal a regulated oligomerization mechanism of the PlexinD1/GIPC/myosin VI complex" for consideration by *eLife*. Your article has been favorably evaluated by Philip Cole (Senior Editor) and three reviewers, one of whom, Suzanne Pfeffer (Reviewer #1), is a member of our Board of Reviewing Editors.

The reviewers have discussed the reviews with one another and the Reviewing Editor has drafted this decision to help you prepare a revised submission.

GIPCs comprise a family of adaptor proteins that link semaphorin signalling via plexin receptors to cellular transport via myosin VI motors. Previous work has elucidated structures of the cytosolic portions of plexins, but the molecular mechanisms how GIPCs couple plexin receptors with myosin motors have remained enigmatic. Here, Shang et al. present a comprehensive structural analysis revealing the molecular mechanisms of GIPC auto-inhibition and activation. In particular, they report the first full length crystal structure of a GIPC showing a domain swapped dimer that is auto-inhibited by intra- and intermolecular interactions. The structure of GIPC bound to a cytosolic plexinD1 fragment reveals the mechanism how auto-inhibitory restraints in GIPC are released. Furthermore, the structure of the GIPC-GH2 domain in complex with a helical fragment of myosin VI uncovers an unexpected mechanism how GIPC clusters myosin VI motors. The structural work is supported by some rather basic biochemical and cell-based experiments. The current manuscript describes a spectacular, high quality structural analysis that could easily be included in several publications. The structure-based functional analyses are not as exhaustive as the structural work, but essentially confirm the main implications of the structures.

1) The reviewers agree that the text can be improved.

2) The authors must show a gel to determine if degradation has taken place in the crystals (subsection “Structure of the PlexinD1/GIPC1 complex”, second paragraph).

3) General electron density must be shown throughout (probably in supplementary information).

4) And quantitation of the micrographs is required.

*Reviewer #1:*

1) Subsection “Structure of the PlexinD1/GIPC1 complex”, second paragraph. It should not be hard to rule out "degradation" in the crystal to make this conclusion stronger. Please check this by running a gel.

2) The light microscopy needs quantitation – number of puncta per cell, number cells counted. If staining is diffuse (not diffused) that can also be represented as overall fluorescence intensity per micron squared or other metric.

3) The Introduction is clearly written but please improve the discussion to explain the order of events that might lead to MyoVI recruitment. (1, ligand binding, dimerization etc.). Explain the events that would uninhibit GIPC1 etc. Propose models for how this might work, to guide next experiments. At the end of the Introduction, please spell out explicitly which partial structures were already determined and how this study builds on those structures. While mentioned in the text, this will help the reader better appreciate how this work builds on previous work of others.

4) There are a large number of acronyms used throughout the text – where possible, please refer to domains by including the protein from which they derive. Otherwise the text is very challenging for those not studying these particular proteins. Example, the first sentence of the subsection “Interactions in the two interfaces between the GH2 and HCBD domains” should read GIPC1 GH2 with MyoVI HCBD; subsection “Structure and oligomerization of the GH2/myosin VI complex”, first paragraph – at least refer to "HCBD" so the reader knows where to look. If possible, add name of protein above bars in Figure 1–Figure 3 to help the reader.

5) Saponin is not saponine.

*Reviewer #2:*

1) One claim in the Abstract and the work, that I worry about is that whilst a domain swapped dimer is undoubtedly present, whether the inhibition is in trans or cis is dependent on modelling of a linker segment invisible in the electron density and therefore I would say precarious. The manuscript certainly needs to point this out and show this and I think present the other obvious possibility i.e. be fairer to an alternative model, which does not detract from the work nor its meaning or 'publishability 'in *eLife*.

2) Subsection “Structure of GIPC1 in the dimeric and autoinhibited state”, third paragraph – inhibition in trans. I think one suggestion of a distance as determining whether inhibition is cis or trans is not acceptable. The linker could go 50Å or so if fully extended so both the 20Å and 40Å possibilities work. I don't think it matters for the impact of the paper or the biology which is true in vivo, or indeed if it is the same as the situation in the crystal but I feel it should be pointed out as a possibility. It would be good to show some support for the model. The phrase "autoinhibited conformation that hinders binding to both PBM-containing cargo proteins and myo6" should be supported with experimental data – pull downs to show weaker binding would be obvious and would improve this section greatly – or soften the text.

3) Subsection “Structure of the PlexinD1/GIPC1 complex”, third paragraph. Again, the phrase "supporting the notion that…" is not really acceptable. Showing the competition should be relatively trivial in a pull-down experiment and this could and should be done I think.

4) Subsection “Structure of the PlexinD1/GIPC1 complex”, last paragraph. Figure 2 panels are very small and cluttered and difficult to understand. Please make bigger and/or simpler.

5) Subsection “Structure and oligomerization of the GH2/myosin VI complex”, second paragraph. Why do the authors believe that myo6 and GIPC polymerisation is needed to cluster receptors in CCPs in initial stages of CME? Which receptors? Clathrin and its associated adapters, mainly AP2, do this efficiently in vivo and in vitro so why the need to invoke myo6/GIPC in it and in any case this would only work for a very small subset of receptors. I also fail to see how this tallies with quantitative mass spec data such as Borner and Robinson showing that there is hardly any of either GIPC and myo6 in CCPs/CCVs. The long-distance runs of endocytic vesicles over actin presumably occur after CCV uncoating and the clathrin/adapter coat would block this interaction with large actin filaments. I think that this whole section of the model needs rethinking and rewriting.

*Reviewer #3:*

Figure 4 should be quantified from 2-3 repetitions, some quantification of the co-localization in Figure 6 would also be desirable.

Figure 5, dashed lines for GH2, HCBD, linker GH2: If I understand correctly, these lines are based on a single measurement at a low protein concentration (30 µM for GH2, 20 µM for HCBD)? It would be important to determine additional sedimentation values for the isolated GH2, HCBD and linker GH2 domains at the same high protein concentrations as used for the complex to confirm that these proteins do not aggregate on their own at high protein concentrations.

Figure 5—figure supplement 1. I am not an AUC expert, but I found the normalization of the traces to the highest value quite confusing. Why not normalize the traces to the total area under the curves for a better comparison of the fraction size of each peak? When looking at the majority of the traces, discrete s20 species are apparent. Can these be assigned to monomers, dimers, trimers, tetramers and pentamers? Is the weighted average sedimentation coefficient really the best means to analyze AUC runs with such distinct species?

Figure 6: 'In these cells, a fraction of PlexinD1 was incompletely processed and remained in the ER or Golgi apparatus.' What is this assumption based on? Please provide the data.

Generally: I find it important if the method of structure solution (e.g. molecular replacement with what?), the diffraction limit of the crystals and the R_work_/R_free_ are shortly mentioned for every structure referred to in the manuscript to immediately provide a measure of the quality of the structure, without having to refer to the data collection table.

Figure 1: The GIPC structure has several completely new features that should be more thoroughly described and analysed. Please enlarge the figures as much as possible that details can be better recognized. Please provide a topology plot (with all secondary structure elements labelled) of the dimeric GIPC structure to better understand the domain architecture, organization and the domain swap of GIPC. In the supplement, please provide structural comparisons of the GH1 and GH2 domains to related domains in other proteins (e.g. ubiquitin, etc.) to better describe their structural relationships. Also, a one sentence description of the PDZ domain architecture should be included in the text. In the close-ups, please label the indicated residues and indicate more secondary structure element, N and C-termini and use colors that have contrast (for example, a light green secondary structure on a yellow surface is not suited). In printouts, it is generally difficult to recognize much below semi-transparent surfaces, especially in small figures as those in the manuscript. Why not use a proper surface representation with some more contrast and provide additional figures to specifically show residues involved in intra- and intermolecular interactions (ideally with a minimum of additional secondary structure elements in the background – better remove it instead of fading it out). Include properly labelled surface conservation plots of the domains to better visualize which contact areas are conserved and which are not.

Discuss the domain swap: Would a monomeric GIPC be stable or would the removal of one monomer lead to a collapse of the GH1 domain? Is there a structural possibility to envisage a monomeric GIPC, e.g. one without domain swap, for example, when the polypeptide chain is newly synthesized?

Figure 2. Although the structure of the plexin cytoplasmic domain has been described before, it still would be helpful for the casual reader to properly describe the domain architectures of the crystallized constructs here. For example, mention that the GAP domain is constituted from C1 and C2 and has a fold related to RasGAPs. Describe in more detail which structures of plexins have been determined before and how do they compare to the current structure. Add 'mouse PlexinD1' in Figure 2, indicate the boundaries of the crystallized construct below, indicate in Figure 2 which structure is the apo and which the GIPC-bound form (the figure should be self-explanatory without reading the legend). Use separate colors for the C1, RBD, C2 domains, indicate the binding sites for small GTPases (are they accessible in these structures for small G protein binding?), indicate N- and C-termini of the construct. Label the tail segments. In the close-ups of Figure 2, it is hard to recognize anything, the interacting residues in the GH1 and PDZ domain are almost invisible: Remove any non-essential detail (for example, the electron densities in the left figures), better concentrate on visualizing the contacts instead. Indicate relevant additional secondary structure elements, show surface conservation plots in the supplement to visualize whether interacting residues are conserved in plexins and GIPCs. Properly label Figure 2 to indicate which constructs have been superimposed.

From an aesthetical point of view: The secondary structure representations in the Figure 2 close-up look ugly: The two helices of the PlxD1 on the left should be separated by a short loop, and the two strands of the PDZ domain look twisted and weird. Manually re-assign secondary structure elements in pymol and chose another, higher quality strand representation in pymol.

Figure 3: I would recommend to first describe the two assembly interfaces in detail in the text and then refer to the higher order assembly, it is rather confusing and counter-intuitive as it is now. Again, the involved amino acids are difficult to recognize in the semi-transparent surface representation in Figure 3 – either remove the surface or provide two separate figures.

Please provide superpositions of all five GH2-HCBD complex in the ASU in the supplement (including the complex that links the two ASUs). Are they really completely identical? Any structure-based hypothesis why oligomerization in AUC stops at the tetramer/pentamer level? For example, would a related closed tetrameric/pentameric assembly be possible (with some minor reorientation of the assembly sites)?

It is distracting that the mutagenesis data appear at the end of the results, whereas the residues participating in the assembly are described much earlier. The authors may consider including the mutagenesis/AUC data immediately after describing the relevant structural data. For example, some AUC data of Figure 5 would neatly fit below Figure 3. Figure 4, Figure 6 and some AUC data from Figure 5 may be merged in one coherent figure describing the assembly of the Plexin-GIPC-myosin complex.

Figure 5 bottom: Please provide the exact amino acid exchanges of all mutants (numbers) in the figure, not in the legend.

Figure 3—figure supplement 1: Very nice figure, but difficult to recognize on a black background. Why not chose different colors for neighboring ASUs (at least for the colored filament) to better visualize how the continuous filament is formed by adjacent ASUs.

Figure 7: Please label the individual domains of GIPC in the figure (ideally with the corresponding domain colors). Discuss: How do the GIPC-PlxD1 hetero-tetramers relate to the GIPC-myosin oligomer? Do the authors envisage a further 3D cross-linking by the combination of tetramers and oligomers or could the tetramers assemble along the oligomeric GIPC-myosin filament?

Table 1: Please indicate in addition the contents of the ASU, e.g. 1 dimeric GIPC complex, 2 GIPC/plexin dimers, etc.

Table 2: Please provide an additional main figure showing the site of disease-related mutations in the auto-inhibited or liganded GIPC structure and discuss the implications in more detail. This information is most relevant for many medical researchers.

[Editors' note: further revisions were requested prior to acceptance, as described below.]

Thank you for resubmitting your work entitled "Structure analyses reveal a regulated oligomerization mechanism of the PlexinD1/GIPC/myosin VI complex" for further consideration at *eLife*. Your revised article has been favorably evaluated by Philip Cole as the Senior Editor, and Suzanne Prefer as the Reviewing Editor, and is essentially ready to be accepted by *eLife*.

Regarding the text in response to Comment 5 of reviewer #2: for greater precision, we request the following edit to the text, "Previous studies have suggested that GIPC1 and myosin VI contribute to clustering of cell surface receptors into clathrin-coated pits at the initial stage of endocytosis IN CERTAIN CELL TYPES, particularly those with microvilli (Hasson, 2003). In such cells, high-order oligomerization of GIPC1 and myosin VI may make this process more efficient."

---

## [Author Response]

1) The reviewers agree that the text can be improved.

We have made many changes to the text and figures as suggested by the reviewers to enhance the presentation of the work.

2) The authors must show a gel to determine if degradation has taken place in the crystals (subsection “Structure of the PlexinD1/GIPC1 complex”, second paragraph).

As suggested, we have re-grown crystals of the PlexinD1/GIPC1 complex. A large number of crystals were collected, washed in the crystallization buffer and analyzed on SDS-PAGE. The result is now shown as Figure 3—figure supplement 2, demonstrating that the crystals contain intact PlexinD1 and GIPC1. There are a few weak bands besides the strong bands of intact PlexinD1 and GIPC1. These are likely small amounts of impurity or partially degraded proteins, which precipitated during the 3-day incubation in the crystallization drops at room temperature and stuck to crystal surface or “crystallization drop skin” that couldn’t be washed away before the gel analysis. We have added the following sentence to the text “A gel analysis shows that the GIPC1 protein is intact in crystals (Figure 3—figure supplement 2), suggesting that the linker-GH2 domains are present but completely disordered in the PlexinD1/GIPC1 complex structure.”

3) General electron density must be shown throughout (probably in supplementary information).

We have included one figure of electron density map for a key part of each structure reported in the paper. We do not show electron density for the entire structures because it becomes highly cluttered and difficult to interpret. The structure factors and coordinates for all the structures reported in the manuscript have been deposited into the PDB database, and will be released as soon as the paper is accepted. The quality of the structures can be checked thoroughly by interested readers.

*4) And quantitation of the micrographs is required.*

This has been done and included in Figure 8 in the revised manuscript. We counted puncta with Sema3E/PlexinD1, myosin VI and GIPC1 co-localized and area larger than 1 µm^2^ in each group.

*Reviewer #1:*

*1) Subsection “Structure of the PlexinD1/GIPC1 complex”, second paragraph. It should not be hard to rule out "degradation" in the crystal to make this conclusion stronger. Please check this by running a gel.*

See the answer to major point 2 above.

*2) The light microscopy needs quantitation – number of puncta per cell, number cells counted. If staining is diffuse (not diffused) that can also be represented as overall fluorescence intensity per micron squared or other metric.*

See the answer to major point 2 above. We realized that the description regarding “diffuse” is vague and difficult to quantify, and decided to use the number of puncta with area larger than 1 µm^2^ as a simple metric for comparing the different groups regarding their ability to form large oligomers in cells. The descriptions regarding “diffuse” are removed.

*3) The Introduction is clearly written but please improve the discussion to explain the order of events that might lead to MyoVI recruitment. (1, ligand binding, dimerization etc.). Explain the events that would uninhibit GIPC1 etc. Propose models for how this might work, to guide next experiments. At the end of the Introduction, please spell out explicitly which partial structures were already determined and how this study builds on those structures. While mentioned in the text, this will help the reader better appreciate how this work builds on previous work of others.*

These are excellent suggestions. We have added the following sentences to the beginning of the last paragraph in the Introduction “Structures of plexins, the PDZ domain of GIPC2 (PDB ID: 3GGE) and the GIPC-binding region in myosin VI have been reported previously (Bell et al., 2011; He et al., 2016; He et al., 2009; Tong et al., 2009; Wang et al., 2013; Wang et al., 2012; Wollscheid et al., 2016). However, structures of full-length GIPCs and their complexes with either cargos or myosin VI are not available, hindering our understanding of the mechanisms underlying GIPC functions.”

And the following sentences to the beginning of the Discussion: “Ligand-induced activation of cell surface receptors often leads to their endocytosis. The primary function of GIPCs in this process is to tether the receptors on uncoated endocytic vesicles to myosin VI, which transports the vesicles to fuse with early endosomes (Katoh, 2013).”

*4) There are a large number of acronyms used throughout the text – where possible, please refer to domains by including the protein from which they derive. Otherwise the text is very challenging for those not studying these particular proteins. Example, the first sentence of the subsection “Interactions in the two interfaces between the GH2 and HCBD domains” should read GIPC1 GH2 with MyoVI HCBD; subsection “Structure and oligomerization of the GH2/myosin VI complex”, first paragraph – at least refer to "HCBD" so the reader knows where to look. If possible, add name of protein above bars in Figure 1–Figure 3 to help the reader.*

Done as suggested.

*5) Saponin is not saponine.*

Corrected.

*Reviewer #2:*

*1) One claim in the Abstract and the work, that I worry about is that whilst a domain swapped dimer is undoubtedly present, whether the inhibition is in trans or cis is dependent on modelling of a linker segment invisible in the electron density and therefore I would say precarious. The manuscript certainly needs to point this out and show this and I think present the other obvious possibility i.e. be fairer to an alternative model, which does not detract from the work nor its meaning or 'publishability 'in eLife.*

We agree and have modified the text to make it more clear that the alternative connection is possible: “The alternative connection is less likely but possible, as the gap of ~40 Å in this configuration can be spanned by the 17-residue disordered linker region.”

*2) Subsection “Structure of GIPC1 in the dimeric and autoinhibited state”, third paragraph – inhibition in trans. I think one suggestion of a distance as determining whether inhibition is cis or trans is not acceptable. The linker could go 50Å or so if fully extended so both the 20Å and 40Å possibilities work. I don't think it matters for the impact of the paper or the biology which is true in vivo, or indeed if it is the same as the situation in the crystal but I feel it should be pointed out as a possibility. It would be good to show some support for the model. The phrase "autoinhibited conformation that hinders binding to both PBM-containing cargo proteins and myo6" should be supported with experimental data – pull downs to show weaker binding would be obvious and would improve this section greatly – or soften the text.*

We have modified the text to clearly point out the alternative possibility. Regarding the autoinhibition, the results in Figure 4 (Figure 6 in the revised manuscript) address this question. The bindings between myosin VI and GIPCs are weak, but they are enhanced in the presence of PlexinD1. These results support the notion that PlexinD1 competes with GH2 for binding to the PDZ domain, leading to dislodge of GH2 and enhanced interaction with myosin VI.

*3) Subsection “Structure of the PlexinD1/GIPC1 complex”, third paragraph. Again, the phrase "supporting the notion that…" is not really acceptable. Showing the competition should be relatively trivial in a pull-down experiment and this could and should be done I think.*

See answer to point 2, the results in Figure 4 (Figure 6 in the revised manuscript) address this concern.

*4) Subsection “Structure of the PlexinD1/GIPC1 complex”, last paragraph. Figure 2 panels are very small and cluttered and difficult to understand. Please make bigger and/or simpler.*

We agree and have split the original Figure 2 into two figures. Figure 2 is now enlarged and shown together with 2E and 2F in new Figure 4.

*5) Subsection “Structure and oligomerization of the GH2/myosin VI complex”, second paragraph. Why do the authors believe that myo6 and GIPC polymerisation is needed to cluster receptors in CCPs in initial stages of CME? Which receptors? Clathrin and its associated adapters, mainly AP2, do this efficiently* in vivo *and in vitro so why the need to invoke myo6/GIPC in it and in any case this would only work for a very small subset of receptors. I also fail to see how this tallies with quantitative mass spec data such as Borner and Robinson showing that there is hardly any of either GIPC and myo6 in CCPs/CCVs. The long-distance runs of endocytic vesicles over actin presumably occur after CCV uncoating and the clathrin/adapter coat would block this interaction with large actin filaments. I think that this whole section of the model needs rethinking and rewriting.*

The review article (Hasson, 2003, cited in the sentence in question) discussed this less well known function of GIPC1 and myosin VI, particularly in microvili of polarized cells. This review article cites a number of papers suggesting this function, in addition to the better-known role of GIPC and myosin VI in transport of uncoated vesicles. We have modified the sentence to make this point more clear. “Previous studies have suggested that GIPC1 and myosin VI contribute to clustering of cell surface receptors into clathrin-coated pits at the initial stage of endocytosis (Hasson, 2003). The high-order oligomerization of GIPC1 and myosin VI may make this process more efficient.”

*Reviewer #3:*

*Figure 4 should be quantified from 2-3 repetitions, some quantification of the co-localization in Figure 6 would also be desirable.*

Quantification of Figure 6 is now included in (new Figure 8). We prefer not to quantify gels, as they show large variation despite that the results are qualitatively robust and reproducible. Figure 11 shows some of the repeats we have done on GIPC1 (A) and GIPC2 (B). Note the G98Y mutant of GIPC2 is equivalent to G114Y of GIPC1. In addition, we did similar experiments using zebra fish GIPC1 and PlexinD1 (C). We tested the two constructs of zebra fish PlexinD1, zfPlxD1152end (with the juxtamembrane segment) and zfPlxD1194end (without the juxtamembrane segment). All the results are similar to those showing in Figure 4 in the original manuscript (Figure 6 in the revised manuscript), with PlexinD1 enhancing the interaction between myosin VI and GIPC.

Author response image 1.**DOI:**
http://dx.doi.org/10.7554/eLife.27322.026

*Figure 5, dashed lines for GH2, HCBD, linker GH2: If I understand correctly, these lines are based on a single measurement at a low protein concentration (30 µM for GH2, 20 µM for HCBD)? It would be important to determine additional sedimentation values for the isolated GH2, HCBD and linker GH2 domains at the same high protein concentrations as used for the complex to confirm that these proteins do not aggregate on their own at high protein concentrations.*

We appreciate the reviewer’s careful examination of this issue, which helped us discover some errors in the figure and improve our data analyses. In original Figure 5—figure supplement 1, the HCBD concentration was labeled “20 µM”, which actually should be 150 µM. Therefore, we actually did do experiments on the HCBD at both low and high concentrations, with the caveat that they were conducted at low-salt (150 mM) and high-salt (500 mM) concentrations respectively. In addition, we found that our initial analyses of the data from the HCBD experiments had some errors, which led to our conclusion as stated in the text “The s_20,w_-value of the HCBD (~5 kDa) at 20 μm was difficult to discern…”. We have corrected the errors by re-doing the analyses, and compared the results for the HCBD at both the low and high concentrations (new Figure 7—figure supplement 1). At the low concentration, the HCBD is a monomer (s_20,w_-value of 0.9). At the high concentration, the HCBD sedimented predominantly as a monomer, but also showed a minor dimeric species (s_20,w_-value of 1.6). There is no obvious higher oligomers formed, and the dimer species is much smaller than the complexes. We now also include the results of the linker-GH2 run at 200 µM, which showed the same s_20,w_-value of 1.4 as at the low concentration, corresponding to a monomer (Figure 7—figure supplement 1). These new analyses and data together show that the linker-GH2 and HCBD do not tend to form large homo-oligomers or aggregate under our experimental conditions. Additionally, we can rule out that the HCBD alone aggregates, based on the fact that it does not aggregate in the presence of the GH2 or linker-GH2 mutants. Although we do not have high-concentration sedimentation information on GH2, we feel it is unlikely to oligomerize on its own, given that the linker-GH2 construct does not oligomerize at high concentrations.

*Figure 5—figure supplement 1. I am not an AUC expert, but I found the normalization of the traces to the highest value quite confusing. Why not normalize the traces to the total area under the curves for a better comparison of the fraction size of each peak? When looking at the majority of the traces, discrete s20 species are apparent. Can these be assigned to monomers, dimers, trimers, tetramers and pentamers? Is the weighted average sedimentation coefficient really the best means to analyze AUC runs with such distinct species?*

We agree with the reviewer regarding normalization, and have revised these figures using the normalization strategy suggested by the reviewer. Regarding the assignment of monomer, etc., that could be done, but would be highly speculative, as these “species” could represent time-averaged sedimentation coefficients of rapidly associating and dissociating complexes, the s-values of which are not represented well by the appearances of the curves. The weighted s-values therefore are an accurate and succinct way to describe the overall behavior of the system.

*Figure 6: 'In these cells, a fraction of PlexinD1 was incompletely processed and remained in the ER or Golgi apparatus.' What is this assumption based on? Please provide the data.*

A figure of immune-staining of PlexinD1 with an anti-PlexinD1 antibody is included in Figure 8—figure supplement 1. It is clear that a portion of PlexinD1 remained in the peri-nuclear region.

*Generally: I find it important if the method of structure solution (e.g. molecular replacement with what?), the diffraction limit of the crystals and the R_work_/R_free_ are shortly mentioned for every structure referred to in the manuscript to immediately provide a measure of the quality of the structure, without having to refer to the data collection table.*

We now mention the diffraction limit of the structures in the text.

*Figure 1: The GIPC structure has several completely new features that should be more thoroughly described and analysed. Please enlarge the figures as much as possible that details can be better recognized. Please provide a topology plot (with all secondary structure elements labelled) of the dimeric GIPC structure to better understand the domain architecture, organization and the domain swap of GIPC. In the supplement, please provide structural comparisons of the GH1 and GH2 domains to related domains in other proteins (e.g. ubiquitin, etc.) to better describe their structural relationships. Also, a one sentence description of the PDZ domain architecture should be included in the text. In the close-ups, please label the indicated residues and indicate more secondary structure element, N and C-termini and use colors that have contrast (for example, a light green secondary structure on a yellow surface is not suited). In printouts, it is generally difficult to recognize much below semi-transparent surfaces, especially in small figures as those in the manuscript. Why not use a proper surface representation with some more contrast and provide additional figures to specifically show residues involved in intra- and intermolecular interactions (ideally with a minimum of additional secondary structure elements in the background – better remove it instead of fading it out). Include properly labelled surface conservation plots of the domains to better visualize which contact areas are conserved and which are not.*

These are great suggestions. Accordingly, we have implemented the following changes in the revised manuscript: (1) Split Figure 1 into two. In new Figure 1, we now include expanded views of the GH1 domain-swapped dimer, the PDZ domain and the linker-GH2 domain. These new panels illustrate the architecture of each domain clearly. We attempted to draw a topology model, which turned out to be quite confusing and therefore decided not to show it. (2) New Figure 2 now show enlarged view of the auto-inhibitory interactions mediated by the linker-GH2 domains. (3) Included a sentence to describe the overall structure of the PDZ domain. (4) Included more labels such as for N- and C-termini of domains.

*Discuss the domain swap: Would a monomeric GIPC be stable or would the removal of one monomer lead to a collapse of the GH1 domain? Is there a structural possibility to envisage a monomeric GIPC, e.g. one without domain swap, for example, when the polypeptide chain is newly synthesized?*

The description regarding this point is in the Methods section of the original manuscript. When expressed in *E. coli*, monomeric GIPCs could form, but was unstable. It is hard to speculate whether there is a portion of monomers of GIPCs in eukaryotic cells, but the dimeric nature of GIPCs has been well established by previous studies as cited in the manuscript.

*Figure 2. Although the structure of the plexin cytoplasmic domain has been described before, it still would be helpful for the casual reader to properly describe the domain architectures of the crystallized constructs here. For example, mention that the GAP domain is constituted from C1 and C2 and has a fold related to RasGAPs. Describe in more detail which structures of plexins have been determined before and how do they compare to the current structure. Add 'mouse PlexinD1' in Figure 2, indicate the boundaries of the crystallized construct below, indicate in Figure 2 which structure is the apo and which the GIPC-bound form (the figure should be self-explanatory without reading the legend). Use separate colors for the C1, RBD, C2 domains, indicate the binding sites for small GTPases (are they accessible in these structures for small G protein binding?), indicate N- and C-termini of the construct. Label the tail segments. In the close-ups of Figure 2, it is hard to recognize anything, the interacting residues in the GH1 and PDZ domain are almost invisible: Remove any non-essential detail (for example, the electron densities in the left figures), better concentrate on visualizing the contacts instead. Indicate relevant additional secondary structure elements, show surface conservation plots in the supplement to visualize whether interacting residues are conserved in plexins and GIPCs. Properly label Figure 2 to indicate which constructs have been superimposed.*

*From an aesthetical point of view: The secondary structure representations in the Figure 2 close-up look ugly: The two helices of the PlxD1 on the left should be separated by a short loop, and the two strands of the PDZ domain look twisted and weird. Manually re-assign secondary structure elements in pymol and chose another, higher quality strand representation in pymol.*

These are great suggestions and we have implemented most of them. Particularly, we have split Figure 2 into two figures (now Figure 3 and Figure 4), and made most of the panels bigger. The PBM/GIPC1 interface is now illustrated with two panels, one with surface representation and another with a detailed view of interacting residues.

*Figure 3: I would recommend to first describe the two assembly interfaces in detail in the text and then refer to the higher order assembly, it is rather confusing and counter-intuitive as it is now. Again, the involved amino acids are difficult to recognize in the semi-transparent surface representation in Figure 3 – either remove the surface or provide two separate figures.*

*Please provide superpositions of all five GH2-HCBD complex in the ASU in the supplement (including the complex that links the two ASUs). Are they really completely identical? Any structure-based hypothesis why oligomerization in AUC stops at the tetramer/pentamer level? For example, would a related closed tetrameric/pentameric assembly be possible (with some minor reorientation of the assembly sites)?*

*It is distracting that the mutagenesis data appear at the end of the results, whereas the residues participating in the assembly are described much earlier. The authors may consider including the mutagenesis/AUC data immediately after describing the relevant structural data. For example, some AUC data of Figure 5 would neatly fit below Figure 3. Figure 4, Figure 6 and some AUC data from Figure 5 may be merged in one coherent figure describing the assembly of the Plexin-GIPC-myosin complex.*

We have enlarged the panels in Figure 3 (new Figure 5) showing the interfaces to make them more clear. We have included a supplemental figure to show that both interfaces I and II made by the different promoters in the crystals are essentially the same in both GIPC1-GH2/myosin VI and GIPC2-GH2/myosin VI structures (Figure 5—figure supplement 3). As mentioned in the original manuscript, pressure induced by high centrifugation force has been shown to inhibit long, linear polymers, which may explain the lack of large oligomers in the AUC experiments. Due to the complexity of the paper, both ways of arranging the data and text have their pros and cons, we prefer to keep as it is.

*Figure 5 bottom: Please provide the exact amino acid exchanges of all mutants (numbers) in the figure, not in the legend.*

Done.

*Figure 3—figure supplement 1: Very nice figure, but difficult to recognize on a black background. Why not chose different colors for neighboring ASUs (at least for the colored filament) to better visualize how the continuous filament is formed by adjacent ASUs.*

We have changed the background to white. We prefer to use different colors for the GIPC-GH2 and myosin VI-HCBD to illustrate the alternating pattern. Instead of using different colors for different ASUs, we now label the three ASUs that form the colored filament to highlight how they connect to each other.

*Figure 7: Please label the individual domains of GIPC in the figure (ideally with the corresponding domain colors). Discuss: How do the GIPC-PlxD1 hetero-tetramers relate to the GIPC-myosin oligomer? Do the authors envisage a further 3D cross-linking by the combination of tetramers and oligomers or could the tetramers assemble along the oligomeric GIPC-myosin filament?*

We have labelled the domains in GIPC in Figure 7 as suggested. Due to the flexibility of the linkers in both GIPC and myosin VI, both types of interactions are geometrically possible. The Discussion section of the manuscript mentioned the 2D cross-linking possibility (3D if one considers the longitudinal plexin/GIPC/myosin interactions as another dimension). We did not get into details of this point because the actual size and organization of oligomers likely vary depending on the shape of receptor dimers and curvature of vesicles.

*Table 1: Please indicate in addition the contents of the ASU, e.g. 1 dimeric GIPC complex, 2 GIPC/plexin dimers, etc.*

Done.

*Table 2: Please provide an additional main figure showing the site of disease-related mutations in the auto-inhibited or liganded GIPC structure and discuss the implications in more detail. This information is most relevant for many medical researchers.*

Great suggestion. A new figure (Figure 10) is now included.